# Thermoresponsive nanoemulsion-based gel synthesized through a low-energy process

Seyed Meysam Hashemnejad [1], Abu Zayed Md Badruddoza [1], Brady Zarket[2], Carlos Ricardo Castaneda [1] & Patrick S. Doyle [1]

Thermoresponsive nanoemulsions find utility in applications ranging from food to pharmaceuticals to consumer products. Prior systems have found limited translation to applications due to cytotoxicity of the compositions and/or difficulties in scaling-up the process. Here, we report a route to thermally gel an oil-in-water nanoemulsion using a small amount of FDA-approved amphiphilic triblock Pluronic copolymers which act as gelling agents. At ambient temperature the suspension displays liquid-like behavior, and quickly becomes an elastic gel at elevated temperatures. We propose a gelation mechanism triggered by synergistic action of thermally-induced adsorption of Pluronic copolymers onto the droplet interface and an increased micelle concentration in the aqueous solution. We demonstrate that the system's properties can be tuned via many factors and report their rheological properties. The nanoemulsions are prepared using a low-energy process which offers an efficient route to scale-up. The nanoemulsion formulations are well-suited for use in cosmetics and pharmaceutical applications.

[1] Department of Chemical Engineering, Massachusetts Institute of Technology, Cambridge, MA 02142, USA. [2] L'Oréal Research and Innovation, Clark, NJ 07066, USA. Correspondence and requests for materials should be addressed to P.S.D. (email: pdoyle@mit.edu)

ydrogels are a unique class of materials with tunable physicochemical and biological properties that are well-suited for variety of biomedical applications[1–3]. Their crosslinked three-dimensional network structures contain a significant amount of water, yet still behave like an elastic solid[1,2,4]. Hydrogels are synthesized by chemical (covalent) or physical (noncovalent) crosslinking of natural or synthetic polymers and other small molecules[4–6]. Physically crosslinked hydrogels, which avoid the use of toxic chemical crosslinking reagents, are of the great interest due to inherent reversibility and dynamics of their physical interactions. The dynamic nature of a physically self-assembled gel network can prevent permanent dissociation of the microstructure and promotes self-healing. The self-healing properties of hydrogels have been shown in different physical crosslinking strategies, such as hydrophobic interactions[7,8], hydrogen bonding[9,10], electrostatic interactions[11,12], and host-guest interactions[13,14]. Physically self-assembled hydrogels can also be formed in response to an external stimuli, such as solvent properties[6], pH[10], redox[14], light[15], and temperature[16]. Among these stimuli-responsive systems, thermal-responsive hydrogels that undergo a sol-to-gel transition at ambient to physiological temperature are highly desirable for biological applications. Most of the thermogelling polymer hydrogels reported in the literature are based on synthetic amphiphilic block copolymers or biopolymers with temperature-sensitive moieties[16–18]. Ideally, the thermogelling formed hydrogels should possess mechanical properties appropriate to the targeted application. For example, if used as an injectable vehicle or in topical formulations it is desirable to have substantial decreases in viscosity as they flow (shear-thinning), rapidly forming a gel at elevated temperature and fast self-healing properties[19,20].

Organohydrogels containing hydrophilic and lipophilic domains have gained great interest in recent years[21–25]. Emulsion-based organohydrogels are highly desirable as they allow for the facile loading of bioactive compounds[24–26]. The encapsulation procedures require only mild dissolution processes, which eliminate any additional chemical modification of bioactive components. The immiscible domains (e.g. oil and water) can be achieved through emulsification methods, which are broadly classified as either high- or low-energy processes[27–30]. Nanoscale emulsions with sizes on the order of 100 nm (so-called nanoemulsions) provide an efficient and facile approach to encapsulating both polar and non-polar functional biomolecules for co-delivery of dissimilar therapeutics[31–33]. Multiple strategies have been employed to obtain thermogelling emulsion-based organohydrogels which are mostly based on entrapping the dispersed phase via thermally gelling the continuous phase[34–37]. Alternatively, we recently developed a method in which the nanoemulsion droplets themselves are physically crosslinked by a thermoresponsive gelator molecule which can bridge nanoemulsion droplets to induce a gel-like material at elevated temperatures[25]. Despite the advances in the synthesis methodologies of thermogelling nanoemulsions, prior approaches are not suitable for practical applications due to the cytotoxicity of the compositions and/or difficulties in scaling-up the process.

Here, we report a unique class of thermogelling organohydrogels which are based on a unique gelation mechanism. The key components in the oil-in-water thermogelling nanoemulsion are chosen among the U.S. Food and Drug Administration's (FDA) approved compounds, which provides a compatible environment for incorporating pharmaceutical agents. In addition, the nanoemulsion suspensions are formed through a low energy process, which facilitates large-scale production[28,30]. Compared with microemulsions, which are a thermodynamically stable system, nanoemulsions are metastable and considered to be an ideal system in many

applications as the droplets remain stable when subject to environmental changes, such as temperature change or dilution[38,39]. The key concept in the present system is the exploitation of hydrophobic interactions between the amphiphilic triblock copolymers and nanoemulsion droplets which work cooperatively in inducing the gel formation. The gelation is thermally triggered as the copolymer is both adsorbed on the surface of the nanoemulsion oil droplets and forms micelles in solution. Unlike our prior work[25], here the copolymer gelators contain hydrophobic groups in the central region of the molecules and do not directly bridge the droplets. Support of the proposed gelation mechanism is provided by calorimetry experiments, rheological measurements, confocal microscopy, and small angle X-ray scattering. The resulting organohydrogels exhibit unique rheological properties. Specifically, they display shear-thinning behavior in both liquid and gel states, spontaneously form a gel at elevated temperatures, and rapidly self-recover following the cessation of applied large stress. The biocompatibility of the ingredients used in these organohydrogels and tunable mechanical properties of the obtained gels make them a promising thermogelling system for applications such as transdermal medications or cosmetics.

## Results

**Thermogelling nanoemulsion formation.** Here we report a thermogelling nanoemulsion platform by the synergistic interplay between low concentrations of amphiphilic triblock copolymers (4.7% wt.) acting as gelling agents and oil nanodroplets. The triblock copolymers used in this work, commercially known as Pluronics or Poloxamer, (F127/F68: 6/1 g/g) are designated as biocompatible compounds by the FDA with a low degree of cytotoxicity[40]. We chose isopropyl myristate (20% wt.) as a canonical oil, as it has been used both in topical formulations and in transdermal drug delivery due to its therapeutic function and its ability to solubilize large amounts of lipophilic drug molecules[41]. To emulsify the disperse oil phase in the aqueous phase, a mixture of food grade nonionic surfactants, polysorbate (Tween 80) and sorbitan oleate (Span 80), was used. The oil-in-water nanoemulsions were produced using the low-energy phase inversion composition method, also known as emulsion phase inversion (Fig. 1a)[28,30]. This process involves the addition of water into a stirred dispersed phase (oil, surfactant, and cosurfactant) at room temperature. The optimal surfactant HLB (hydrophilic-lipophilic balance) was considered to minimize the droplet size (Supplementary Fig. 1). Poly(ethylene glycol), PEG 400 (5% wt.), was used as a cosurfactant to further reduce the oil droplet size (Supplementary Fig. 1). The formed nanoemulsion exhibited a relatively long-term stability in a diluted state and at an elevated temperature (Supplementary Fig. 2a), which is a signature of metastable suspensions with kinetically stable droplets. In contrast to energy-intensive emulsification processes, such as high pressure homogenization and ultrasonication, the low-energy method employed here only requires simple mixing using a magnetic stirrer. This synthesis method also offers the feasibility of large-scale production of the formulation and batches up to 0.5 kg have been produced. After nanoemulsion preparation (with droplet size of 53 nm, Polydispersity index, PDI, of 0.12), a Pluronic solution was dispersed into the suspension to render it thermally responsive (Fig. 1b). A mixture of Pluronic copolymers (F127/F68: 6/1 g/g) with a final concentration of 4.7% wt. was used to tune the gelation temperature. The formed thermogelling nanoemulsion system also offers a long shelf-life and remained stable for a period of at least one year (Supplementary Fig. 2b).

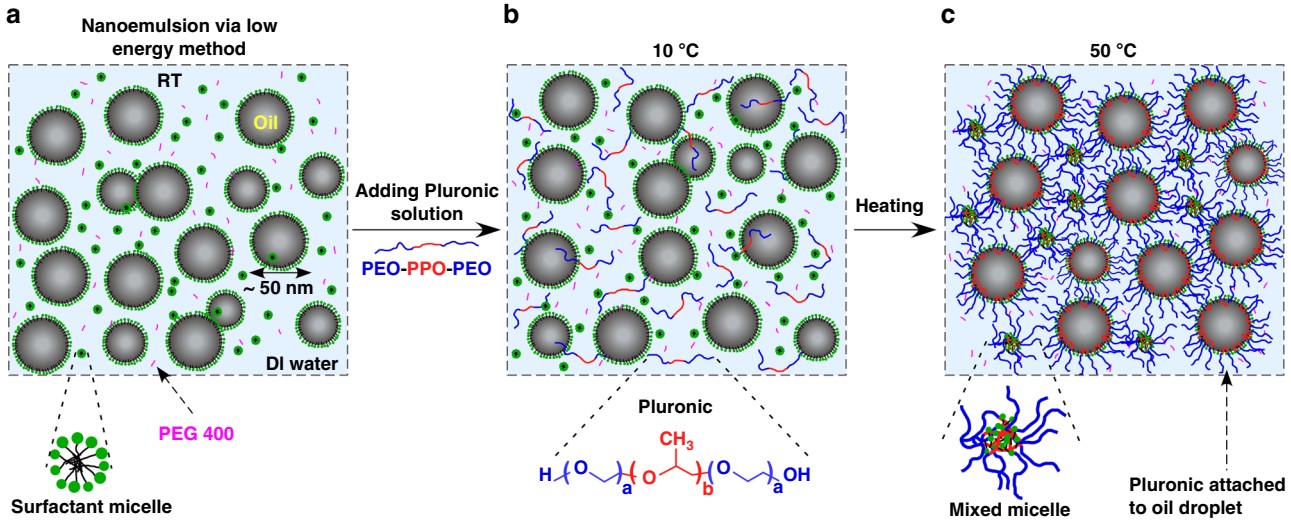

**Fig. 1** Schematic diagram of the nanoemulsion system and hypothesized gelation mechanism. **a** Formation of the oil-in-water nanoemulsion prepared using a low energy process containing oil, surfactant (Tween 80 and Span 80, HLB = 13) and PEG 400 as a cosurfactant. **b** Addition of mixed Pluronic solution into the nanoemulsion suspension to impart thermoresponsive behavior. Final concentration of Pluronic is 4.7% wt. **c** Gel formation at 50 °C through adsorbing midblock of Pluronic onto the droplet interface and mixed micelles formation between Pluronic and excess emulsifiers in the aqueous solution

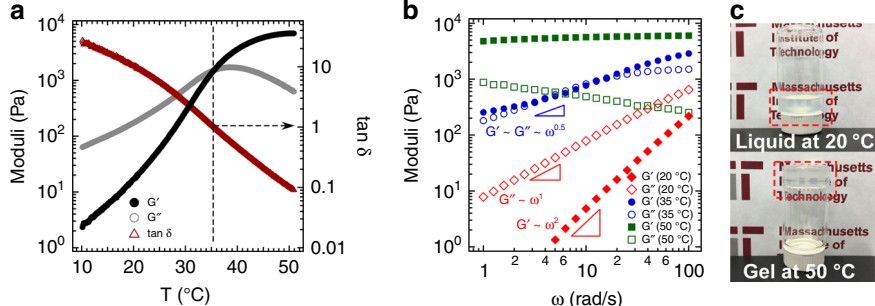

**Fig. 2** Rheological behavior of thermogelling nanoemulsion-based gel. The canonical system contains 20% isopropyl myristate as the oil phase, 20% surfactant as an emulsifier (a mixture of Tween 80 and Span 80 with HLB = 13), 5% PEG 400, 4.7% Pluronic (F127/F68: 6/1 g/g), and 50.3% DI water. All concentrations are in mass percentage. The droplet volume fraction, $\phi$, is 0.24, and the hydrodynamic oil droplet diameter is 53 nm. **a** Evolution of the dynamic moduli in a small amplitude oscillatory shear temperature ramp experiment from 10 °C to 50 °C with a ramp rate of 2 °C/min. The dashed line indicates where $G'$ and $G''$ intersect. **b** Linear viscoelastic spectra at different temperatures. **c** Optical images of inverted glass vials containing sample below (20 °C) and above (50 °C) the gelation temperature

**Sol-to-gel transition of a nanoemulsion**. The sol-to-gel transition of our thermogelling nanoemulsion is displayed in Fig. 2. Evolution of the gel formation as a function of temperature was captured in small amplitude oscillatory shear rheology and vial inversion tests. Temperature ramp measurements at a fixed frequency (Fig. 2a) show that both elastic ($G'$) and loss ($G''$) moduli increase as a function of temperature starting from 10 °C and crossover each other at a critical gel point ($T_{gel} = 35$ °C). Above the gel point, $G'$ exceeds $G''$ and at 50 °C the elastic modulus eventually becomes about one order of magnitude higher than the loss modulus. Frequency sweep tests were conducted at temperatures corresponding to the liquid state, critical gel point, and gel state, Fig. 2b. At low temperature (20 °C), where $G'' \gg G'$, $G'$ ($\omega$) $\sim \omega^2$ and $G''$ ($\omega$) $\sim \omega^1$, which indicate liquid-like behavior. However, at the critical gel point, the moduli display identical power-law scaling with frequency for over two decades of frequency ($G'$ ($\omega$) $\sim G''$ ($\omega$) $\sim \omega^n$, here with $n = 0.5$) which is consistent with the Winter-Chambon criteria for gelation[42]. A similar power exponent has been found in many chemical and physical gels[42,43]. The frequency dependence of the shear moduli above the gel point continues to decrease for both the elastic and loss

components. In the gel state (50 °C, Fig. 2b), $G'$ becomes relatively frequency independent ($G'$ ($\omega$) $\sim \omega^0$) and the power exponent of $G''$ is a negative number, similar to a Maxwell model for non-Newtonian viscoelastic materials[44]. We note that this high temperature elastic modulus is comparable to that found in our prior work, which used bridging gelators to induce nanoemulsion gelation[25] and is stronger than elastic moduli reported for other liquid-liquid emulsion systems[45–49]. Gel formation was also verified in a vial inversion test. Figure 2c displays the optical images of the thermogelling nanoemulsions at liquid (20 °C) and gel (50 °C) states in inverted 20 mL glass vials. In the liquid state the material is able to flow, whereas at gel state the self-assembled nanoemulsions can hold their own weight. We also visually observe that turbidity of the nanoemulsion solution remains unchanged upon gelation—both liquid and gel states are translucent.

**Thermal gelation mechanism and microstructure**. Control experiments were performed to verify the essential role of the nanoemulsion droplets in the thermogelling behavior of our system (Supplementary Fig. 3). The copolymers used in our system are

well-characterized thermal gelators in of themselves, however, only at much larger concentrations[50–52]. Pluronics are amphiphilic triblock copolymers consisting of poly(ethylene oxide)-poly (propylene oxide)-poly(ethylene oxide) (PEO–PPO–PEO). As temperature increases, midblock segments become more hydrophobic, while PEO segments are hydrated over a wide range of temperatures. The temperature-dependent solubility of PPO in water promotes micelle formation. At high enough concentration, these spherical micelles construct a packed microstructure. It has been postulated that overlap of the PEO chains in adjacent micelle coronas gives rise to gelation[53]. Temperature ramp measurements of the mixed Pluronic solution used in this work indicated that the minimum polymer concentration needed for sol-to-gel transition to occur is 13% (Supplementary Fig. 3a) and in a 15% Pluronic solution only a weak gel can be obtained. Additionally, dynamic rheological experiments were conducted on our canonical composition without the presence of the oil (Supplementary Fig. 3c–e) and no gel formation was observed over the temperature range 10–60 °C.

The above control experiments emphasize that the presence and function of the oil phase is critical in obtaining a thermogelling behavior at these very low Pluronic concentrations. We hypothesize that thermally induced gelling in our system occurs by a synergistic effect between the Pluronic copolymers and nanoemulsion droplets (Fig. 1c). There are multiple hydrophobic sites that may affect the self-assembly of Pluronic molecules as temperature rises. These include nanoemulsion oil droplets as well as the hydrophobic core of Tween 80 and Span 80 micelles. As the temperature increases, PPO segments of Pluronic adsorb to the oil interface as they become more hydrophobic. The adsorbed layer of Pluronics onto the oil interface alters the effective volume fraction ($\phi_{eff}$) of the oil droplets[54]. The effective volume fraction of the Pluronic-coated hairy droplets can be estimated using the relationship derived for sterically stabilized dispersions: $\phi_{eff} = \phi\,(\delta/r)^3$, where $\delta$ is the droplet radius with adsorbed layer of Pluronic, $r$ is the initial nanoemulsion radius, and $\phi$ is the true volume fraction. We assume that the oil droplets are mostly coated with F127, as PPO/PEO molar ratio is higher in F127 compared with F68 (PPO/PEO ratios are 0.33 and 0.19 for F127 and F68, respectively) and molar concentration of F127 is as four times as F68. Assuming an adsorbed layer thickness of ≈5.5 nm[55], the effective volume fraction of the oil phase can be increased from 0.24 to 0.43 upon adsorption of the Pluronic. While this is a significant increase in the effective volume fraction, it is still below the random close-packed limit for

monodisperse hard spheres ($\phi = 0.64$). Excess Pluronic copolymers can also form mixed micelles in the aqueous phase, which increases the micelle concentration in the solution (Fig. 1c)[56]. Akin to the jamming-induced gelation in pure Pluronic systems, we hypothesize that the packing of the mixed-micelles with the hairy nanodroplets leads to the observed thermogelation in our system. The fact that there is no gel transition in the absence of oil droplets suggests that the adsorption of Pluronic copolymers onto the droplet interface is a key step, which gives rise to the thermogelling behavior of our system. Additionally, the confocal images captured above the gelation point support the proposed gelation mechanism where the oil droplets are distributed randomly throughout the microstructure (Supplementary Fig. 4). Furthermore, the desmeared small angle X-ray scattering spectra of the thermogelling nanoemulsion nearly overlap at liquid and gel states. This confirms that at gel state no clusters form and the oil droplet size remains intact at an elevated temperature. (Supplementary Fig. 5).

To corroborate our hypothesis regarding the underlying thermogelling mechanism of our system, we performed microcalorimetry measurements (Fig. 3). To find the critical micellization temperature of the Pluronic solution used here (F127/F68), micro-DSC measurements were conducted on different concentrations of the aqueous Pluronic solutions as references (Fig. 3a). The thermograms of Pluronic solutions give rise to an endothermic peak as a result of heating[51,56]. As temperature increases, PPO segments become dehydrated and less polar which eventually aggregate to form micelles resulting in an endothermic peak in the micro-DSC. The micellization process depends on the concentration and chain length of the copolymers. For example, peak temperature decreases from 29 to 23 °C as the Pluronic concentration increases from 1.2 to 7.8%. To evaluate the adsorption of the midblock (PPO) in the Pluronic onto hydrophobic oil interfaces during ramping temperature experiments, the thermogelling nanoemulsion was diluted to decrease the viscosity of the solution sufficiently to load the samples for micro-DSC measurements and to avoid saturating the maximum detection range of the instrument. Nanoemulsions were centrifuged (using centrifugal filters) to filter the oil droplets from the continuous phase and were separately analyzed in DSC experiments. Figure 3b displays DSC traces for the neat Pluronic solution, the subnatant solution (continuous phase), and the washed nanoemulsion (oil) with a similar Pluronic concentration. Interestingly, the endothermic transition peak broadens in the presence of oil droplets and the onset of the peak significantly

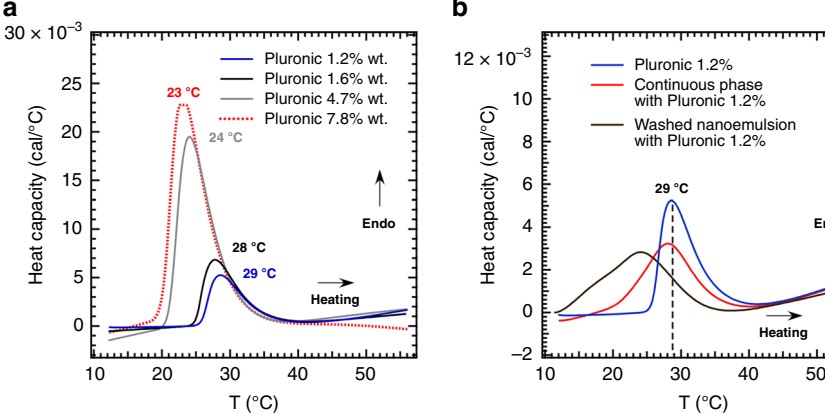

**Fig. 3** Micro-DSC signals after baseline subtraction. DSC traces of, **a** aqueous solutions of F127/F68 (6/1 g/g) at different copolymer concentrations, **b** of Pluronic solution, the washed nanoemulsion (oil 0.4% wt.), and the subnatant solution (continuous phase) with a similar Pluronic concentration of 1.2%. Nanoemulsion suspensions were centrifuged to filter oil droplets (supernatant) from aqueous phase (subnatant) before adding Pluronic solution. The dashed line in b marks the endothermic peak for the 1.2% Pluronic solution

shifts 13 °C to a lower temperature. As temperature increases in the presence of the oil droplets, hydrophobic segments of Pluronic copolymers (PPO groups) are adsorbed onto the oil droplets and results in an endothermic peak. This implies that Pluronic and nanoemulsion droplets interact strongly with each other. In addition, increased micelle formation occurs in the continuous phase. The endothermic transition peak (red curve in Fig. 3b) was centered at 29 °C, similar to the neat Pluronic solution; however, the onset of the endothermic peak shifted 5 °C to a lower temperature. This early onset is likely due to the mixed micelle formation (as shown in Fig. 1c) between Pluronic and free emulsifier molecules (Tween 80 or Span 80) present in the aqueous phase. A similar micro-DSC behavior has been reported for a Pluronic solution in the presence of non-ionic surfactants[56]. Our DSC data suggests that the endothermic adsorption process of Pluronic onto the oil droplet interface is more favourable and dominant as compared with micellization of Pluronic copolymers. Further, unlike the concentrated pure Pluronic solution in which shear moduli sharply increase at the gel point (Supplementary Fig. 3a), the growth of $G'$ and $G''$ in our thermogelling nanoemulsion occurs more gradually (Fig. 2a vs Supplementary Fig. 3a). This might be due to the occurrence of multiple temperature-sensitive phenomena, including forming hairy oil droplets and then Pluronic micelles (suggested from broad copolymer adsorption peak in the oil interfaces vs. narrower endothermic peak in pure Pluronic solution).

To understand the origin of the elasticity of our system, we tested the reversibility of the shear moduli in temperature cycle tests (Supplementary Fig. 6a, b, and c). We noticed that the liquid-like behavior at low temperature is not recoverable through the cooling step (at least during the considered time scale). In fact, the structure remains in the gel state (for a prolong period of time), after being subjected to high temperature. These hysteresis effects are similar to the previously reported data on a concentrated pure Pluronic solution[57]. In pure Pluronics, micelles begin to form as temperature increases until they increase in number to be closely packed. Elasticity in the pure Pluronic system is thought to arise from the overlapping micelle shells formed of PEO chains[53]. It has been further suggested that these overlapping PEO chains may serve as transient physical junctions between micelles[57]. In analogy with the pure Pluronic copolymer solution, we speculate that the origin of the elasticity and gel-like response in our thermogelling nanoemulsion is most likely attributed to a sequence of events as temperature is increased: (1) adsorption of the Pluronic to the nanoemulsion, (2) Pluronic micelle formation, and (3) increase in Pluronic micelles until they become close-packed such that there will be overlap of PEO chains amongst the various entities (both micelles and hairy droplets), as depicted in Fig. 1c.

**Effect of oil droplet size in thermogelling behavior of nenoemulsion.** Next, we prepared nanoemulsions with varied droplet diameters to examine the effect of droplet size on the thermogelling behavior of our nanoemulsion system. The preparation was performed in such a way that all chemical compositions are identical, except for the oil droplet sizes. Figure 4 displays the shear moduli as a function of temperature as well as frequency dependency of the moduli at 50 °C for three different droplet sizes (hydrodynamic diameters equal to 53, 72, and 115 nm). All three systems show thermogelling behavior, though with some qualitative changes in rheological behavior and shifts in gelation temperature. For the system with the largest droplet size (115 nm), the $G'$–$G''$ crossover spans from 30 °C to more than 40 °C. At 50 °C, the storage modulus becomes more frequency dependent as droplet size increases. At a fixed dispersed phase volume fraction, increasing the nanoemulsion size from 53 nm to 115 nm reduces the oil surface area by approximately a factor of 2. Additionally, increasing the droplet size (holding the true volume fraction constant) decreases the effective volume fraction for

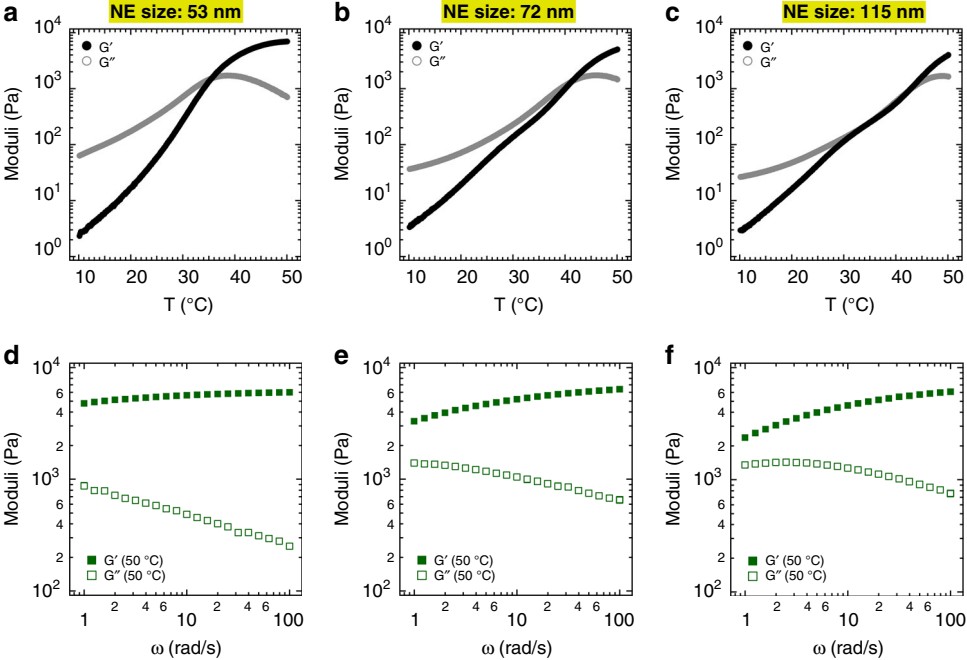

**Fig. 4** Effect of nanoemulsion (NE) droplet size on rheological properties and gelation temperature $T_{gel}$. Shear moduli as a function of temperature and frequency at 50 °C. The top panels show the evolution of the dynamic moduli in a small amplitude oscillatory shear temperature ramp experiment for thermogelling nanoemulsion with hydrodynamic diameters of **a** 53 nm, **b** 72 nm, and **c** 115 nm. Bottom panels **d**–**f** show the frequency sweep test at 50 °C for the same systems (applied strain amplitude is 0.1%). All systems contain the same volume fraction of oil droplets ($\phi = 0.24$) and chemical composition as the canonical system introduced in Fig. 2; 20% isopropyl myristate as an oil phase, 20% surfactant as an emulsifier (a mixture of Tween 80 and Span 80 with HLB = 13), 5% PEG 400, 4.7% Pluronic (F127/F68: 6/1 g/g), and 50.3% DI water. All concentrations are in mass percentage

Pluronic-coated droplets from $\phi_{eff} \approx 0.43$ to 0.32 when increasing the droplets size from 53 to 115 nm. We postulate that decreasing the effective volume fraction of the oil droplet results in the formation of a gel with a less jammed structure at elevated temperatures and hence a decrease in elasticity. Our hypothesis also was confirmed by using different types of Pluronic copolymers with relatively a similar PEO length, and indeed the results indicated a similar thermogelling behavior of the nanoemulsion (Supplementary Fig. 7 and Supplementary table 1).

**Effect of PEG 400 in thermogelling behavior of nanoemulsion.**
Rheological behavior of our thermogelling nanoemulsion can be modulated by other factors in addition to droplet size. To test the effects of the cosurfactant PEG 400 on thermogelling behavior, we varied the concentration of PEG 400 from 5 to 10% (by mass). In order to compare with our canonical gel, these formulations were prepared in a manner such that the thermogelling nanoemulsion would have a similar oil droplet size (53 nm). The viscoelastic moduli as a function of temperature are shown in Fig. 5. Interestingly, increasing the concentration of PEG 400 lowers the gelation temperature. For example, the gel point reduces 10 °C, when PEG 400 is doubled from 5 to 10%. In addition, both the storage and the loss modulus have higher values in the liquid state with an increase of PEG 400 concentration. However, all three systems have similar shear moduli in the gel state (50 °C). The frequency sweep data captured at the gel state also indicate a similar frequency dependence of shear moduli for these thermogelling nanoemulsions. Having higher values of viscoelastic moduli in the liquid state and lower gelation temperatures with increasing concentration of PEG 400, indicate possible hydrogen bonding between hydroxyl groups and ether groups among PEG 400 molecules or between PEG 400 and polyethylene groups in the Pluronics. Prior studies showed that the presence of short polyethylene glycol chains lowers the critical micellization temperature of Pluronic aqueous solutions[58,59].

**Bulk rheological characterization of a thermogelling nanoemulsion.** For a thermogelling material to be used as a topical formulation, certain rheological properties are desirable both below and above the gelation temperature. In this section we discuss these rheological properties of our canonical thermogelling nanoemulsion (Fig. 6). At ambient temperature, flow behavior similar to Newtonian fluids was observed at shear rates lower than 1 s⁻¹; that is, shear stress linearly increases against shear rate and corresponding viscosity is independent of shear rate (a plateau in the viscosity at low shear rates) (Fig. 6a). However, at higher shear rates, the material becomes shear-thinning with a power-law dependence on shear rate, $\eta \sim \dot{\gamma}^m$ with $m = -0.6$. The shear-thinning properties at ambient temperature would allow for easy injection of the material. After the material is deposited on a target surface at an elevated temperature, it should undergo a rapid phase transition in order to preserve the structure. To test the material response, we performed qualitative and quantitative temperature-jump experiments. Figure 6b displays shear moduli in a temperature jump test from 20 to 50 °C (flow behavior as a function of temperature is also presented in Supplementary Fig. 8). Intriguingly, the thermogelling nanoemulsion responds very quickly to a temperature jump and becomes a gel almost instantaneously. In order to demonstrate qualitatively the gelation rate, the room temperature thermogelling nanoemulsion was dripped through a flat-tip 15-gauge needle into a hot water bath held at 50 °C (Fig. 6c and Supplementary Movie 1). For better observation of the process, the oil phase was also loaded with the Nile red (0.05 mg in 1 mL isopropyl myristate). The drops of the initially liquid-like nanoemulsion suspension become a gel as they enter the hot water and form mushroom-like objects. These immersed gel objects retain their shape and persist for several minutes before eventually dissolving and dispersing into the water. This observed behavior could prove useful when considering these materials for applications involving injected drug delivery depots, 3D printing and topical skin formulations.

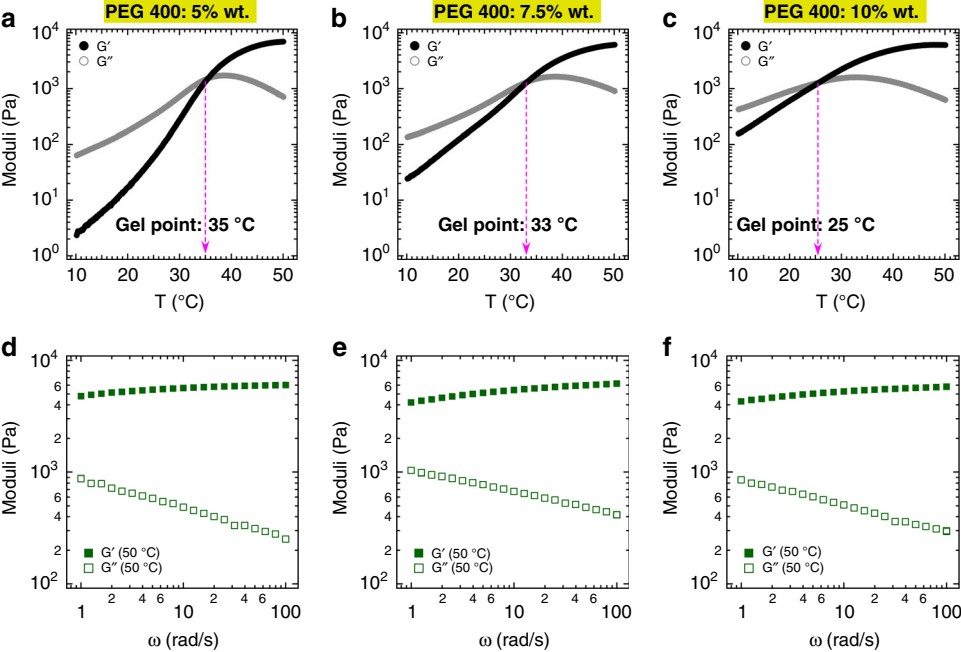

**Fig. 5** Effect of cosurfactant PEG 400 concentration on rheological properties and gelation temperature $T_{gel}$. Shear moduli as a function of temperature and frequency at 50 °C. Nanoemulsion droplet size (hydrodynamic diameter of 53 nm) and volume fraction ($\phi = 0.24$) are kept constant for all formulations. The only difference in the chemical composition of the samples is the amount of added cosurfactant (PEG 400). The top panels **a–c** show the evolution of the dynamic moduli in a small amplitude oscillatory shear temperature ramp experiment and bottom panels **d–f** show frequency sweep test at 50 °C for the same systems. The pink lines with arrows in **a–c** mark the intersection of G′ and G and indicate the gel point

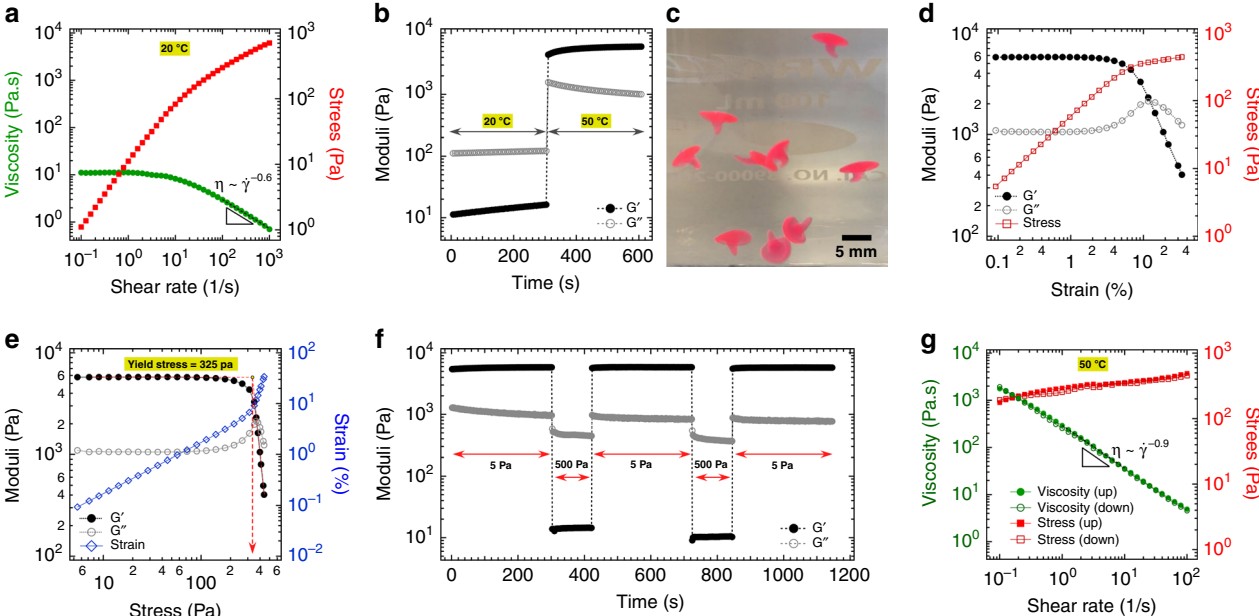

**Fig. 6** Rheological characterization of the thermogelling nanoemulsion below (20 °C) and above (50 °C) the gel point. **a** Steady shear rheology at 20 °C. **b** Viscoelastic moduli in a temperature jump from sol state (20 °C) to gel state (50 °C); ($\gamma_0 = 0.1\%$ and frequency of 20 rad/s). **c** Optical image of mushroom-shaped gel particles formed by dripping a room temperature thermogelling nanoemulsion (using a 15-gauge needle) into a hot water bath at 50 °C. Large amplitude oscillatory shear experiment (frequency of 20 rad/s) plotted vs. **d** strain amplitude, and **e** shear stress. The dashed red line with an arrow in panel **e** indicates the yield stress. **f** Change in the moduli indicating breaking and recovery of the structure under applied stresses of 500 Pa and 5 Pa, respectively ($\gamma_0 = 0.1\%$ and frequency of 20 rad/s). **g** Cyclic flow sweep experiment from low to high shear rate (up) and from high to low shear rate (down) at 50 °C. Dash lines in **a** and **g** are power-law fits to the data

Large amplitude oscillatory shear (LAOS) experiments were conducted to monitor the nonlinear flow behavior of our system[60]. Figure 6d displays shear moduli and stress as a function of applied shear strain amplitude at 50 °C, where the gelled material is highly structured. The storage and loss moduli are independent of strain below a strain amplitude of 2%, indicating a linear viscoelastic regime. At higher applied strain, $G'$ starts to deviate from linearity with a gradual drop in magnitude and above a strain of 10% a sharp decay was observed. Conversely, $G''$ starts to grow beyond the linear regime and passes through a maximum exceeding $G'$ prior to a sharp decay resulting from structural failure of the material. Similar LAOS results have also been reported for emulsion-based gels with a peak in $G''$ at large strain[61]. The transition from linear to nonlinear viscoelastic regimes can also be monitored in a stress-strain curve. Shear stress linearly increases as a function of strain amplitude up to a stress of 110 Pa, beyond which point stress starts to deviate from linearity. In order to better estimate the yield stress, the critical point above which the structured thermogelling nanoemulsion is disrupted and starts to flow, the shear moduli were plotted as a function of stress (Fig. 6e). Two tangent lines were plotted in the linear and the nonlinear regimes of $G'$ and the crossover point determines the yield point. We obtained a yield stress of 325 Pa at the gel state which is comparable to the yield stress in pure Pluronic gels with a much higher copolymer mass fraction of 19%[62].

To test recovery of the elasticity after yielding, low to high oscillatory shear stress was repeatedly applied. Figure 6f displays the moduli as a function of time after two cycles of low (5 Pa for 300 s, below the yield stress) to high (500 Pa for 120 s, above the yield stress) oscillatory stress amplitudes. The application of a shear stress above the material yield stress resulted in moduli indicative of liquid-like behavior ($G'' \gg G'$). However, the structure can recover its full elastic strength after cessation of the applied

large stress. Full recovery of internal structure in the thermogelling nanoemulsion occurs nearly instantly (in less than 10 seconds). In addition, after two cycles of low to high oscillatory stress, the material returned to its initial value of elastic modulus. In order to further investigate the time-dependent rheological properties of our gel, the thixotropic features (breakdown and buildup of the internal structure) were monitored. An established protocol for characterizing the thixotropic behavior was applied in which the shear viscosity and stress is monitored under steady flow cycle of progressively increasing and decreasing shear rate (Fig. 6g). Interestingly, the initially gelled material at $T = 50$ °C shows strong shear-thinning behavior ($\eta \sim \dot{\gamma}^m$ with = −0.9). Additionally, the viscosity is not dependent on time (i.e. not thixotropic), indicated by overlap in viscosity recorded from low to high shear rate, and the reverse. These results also confirm that the full recovery occurs nearly instantaneously after the material is disrupted (as shown in Fig. 6f) to prevent the material from flowing or dripping which is essential in topical formulations.

**Dissolution profile of a hydrophobic active molecule**. One of the potential applications of the formed nanoemulsion is as carrier devices of active compounds for the topical formulations. To demonstrate this application, a model active pharmaceutical ingredient (ibuprofen) was dissolved in the oil phase prior to the nanoemulsion formation and the release profile was studied below and above the gel point of the nanoemulsion. Supplementary Fig. 9 displays the release profiles of the ibuprofen in the dissolution experiments at two different temperature (10 and 37 °C). We noticed that the release profiles are distinctively different at the sol and gel state of the material. At low temperature (where the nanoemulsion has a liquid-like behavior), over 90% of the active material is released within 10 min, whereas at body temperature (where the nanoemulsion forms a gel) it takes about

30 min for the same amount of the ibuprofen to dissolve into the dissolution media.

Overall, our synthesized thermogelling nanoemulsion displays a unique sol-to-gel transition mechanism and offers distinct advantages compared with previously reported thermogelling emulsion-based systems. For example, in our prior work the droplet gelation is thought to occur via droplet bridging induced by a relatively cytotoxic gelator[25]. In contrast, the Pluronic copolymers used here as the thermo-gelator molecule are FDA-approved molecules and have been widely utilized in different biological and pharmaceutical formulations. In addition, the gelation mechanism is different from many other previous reports, in which gelation typically occurs by adding high concentrations of thermogelling polymer (either synthetic or natural) to gel the aqueous phase[34–37]. In these thermogelling emulsion-based systems, the disperse phase has no role in inducing the gelation and is merely loaded material. In contrast, nanoemulsion droplets in our system play a critical role in the thermoresponsive behavior of the formulation by becoming hairy droplets decorated by the Pluronic copolymers at elevated temperatures. Additionally, the nanoscale droplet size and the presence of the gelator copolymer in the aqueous phase render it a long-term stable formulation (over a period of at least one year), and the feasibility of producing the formulation with a low-energy process in bulk makes it suitable for scaled-up commercial applications. To further test the generality of the thermogelling behavior of our nanoemulsion, the oil fraction and Pluronic concentration in the canonical formulation were varied and their rheological properties were characterized (Supplementary Fig. 10 and Supplementary Fig. 11). We found that the thermogelling behavior is robust over a wide range of oil fractions as low as 0.18 (equivalent to 15% wt.) and Pluronic mass fraction as low as 3.9%.

In summary, we synthesized a thermogelling oil-in-water nanoemulsion which contains biocompatible ingredients. Nanoemulsion suspensions are formed through a low-energy method which facilitates large-scale production. A low concentration (4.7% wt.) of Pluronic triblock copolymer was used to gel the nanoemulsion suspension. Micro-DSC data indicate that as temperature increases, nanoemulsion droplets are decorated with Pluronic copolymers such that the midblocks are adsorbed onto the oil-water interfaces while the end blocks remain hydrated in the continuous phase. The decorated nanodroplets and an increased concentration of micelles in the aqueous phase results in a packed gel structure. These nanoemulsion-based gels exhibit unique rheological properties. At ambient temperature, they display shear-thinning behavior, and the solution rapidly gels as temperature is increased. The gel structure is disturbed under applied shear stress above the yielding point and quickly returns to the original state when the applied load is removed. Due to the ease of synthesis, biocompatibility, and the self-healing properties, the introduced thermoresponsive nanoemulsion gel has the potential to serve as an ideal and robust thermogelling vehicle. They warrant further exploration for transdermal and topical formulation application.

## Methods

**Materials**. Isopropyl myristate, Tween 80, Span 80, and Pluronic copolymers (F127 and F68) were purchased from Sigma-Aldrich. Other Pluronic copolymers (F88, F98, and F108) were kindly provided as a gift by BASF Corporation, NJ. Polyethylene glycol 400 (PEG 400) was purchased from TCI Chemical Trading Co., Ltd. All chemicals were used as received.

**Synthesis of thermogelling nanoemulsion**. The oil-in-water nanoemulsions were first prepared, and then followed by the addition of gelator molecules (See Supplementary Fig. 1). Nanoemulsions were prepared by changing the composition in a low-energy methodology at ambient temperature. Deionized (DI) water was

added dropwise into a magnetically stirred mixture of oil, surfactant, and cosurfactant. Subsequent to nanoemulsion formation, a Pluronic solution (F127/F68: 6/1 g/g and with polymer concentration of 23.3%, kept at 4 °C), was added. To thoroughly mix the final solution, it was vortexed for additional 5 s and was kept at 4 °C overnight (to remove air bubbles) prior to experimental analysis. Addition of F68 enhances the flow behavior of the thermogelling nanoemulsion at ambient temperature in comparison with using only F127 as the gelator (Supplementary Fig. 7 vs Fig. 2). The final composition and concentration in our canonical thermogelling nanoemulsion is as follows: isopropyl myristate as an oil (20% wt., equivalent to the volume fraction of 0.24), mixture of Tween 80 and Span 80 (HLB = 13) as surfactant (20% wt.), PEG 400 as cosurfactant (5% wt.), Pluronic, with mass ratio of F127/F68: 6/1, as thermo-gelator molecules (4.7% wt.) and DI water (50.3% wt.). The thermogelling nanoemulsions prepared in this work have a droplet diameter size of 53 ± 2 nm with polydispersity index (PDI) of 0.12, as measured using dynamic light scattering. To monitor the intrinsic stability of the thermogelling nanoemulsion, samples were stored at room temperature and droplet size measurements were monitored for over a period of one year (See Supplementary Fig. 2b).

To prepare thermogelling nanoemulsions with different oil droplet sizes (53, 72, and 115 nm, as presented in Fig. 4), the cosurfactant (PEG 400, total of 5% wt.) was added during and/or after nanoemulsion preparation as follows. It should be noted that PEG 400 functions as a cosurfactant and reduces the oil droplet diameter only if used during emulsification[63]. For example, a nanoemulsion with oil droplet size of 72 nm was prepared by adding 2.5% wt. PEG 400 during and then 2.5% wt. after nanoemulsion formation. Similarly, adding 5% wt. PEG 400 after nanoemulsion formation resulted in a droplet size of 115 nm. In addition, to obtain nanoemulsion suspensions with different concentrations of PEG 400 while having a similar oil droplet diameter (Fig. 5), 5% cosurfactant was used during nanoemulsion formation and additional amount of cosurfactant was incorporated after forming the nanoemulsion.

**Dynamic light scattering**. A NanoBrook 90plus PALS (Brookhaven Instruments Corporation, USA) particle size analyzer was used to measure nanoemulsion droplet sizes and polydispersity index (PDI). Hydrodynamic droplet size was analyzed by dynamic light scattering. Samples were diluted to approximately 0.001 volume fraction. The dilution was performed to eliminate multiple scattering effects arising from components other than the nanodroplets and their inherent effects on the continuous phase viscosity.

**Micro-differential scanning calorimetry**. Micro-differential scanning calorimetry (Micro-DSC) experiments were carried out on a micro-calorimeter (MicroCal Inc., MA, US). The samples were kept at 4 °C for 30 min prior to injecting through an 18-gauge needle into the sample and reference cells. To obtain DI water baselines, an overnight up-and-down temperature scan was performed over a temperature range between 10 and 75 °C at a rate of 1 °C/min (the overall measurement time of the rheology and DSC tests are nearly the same). Subsequent to ensuring bubble free loading in the reference cell and no contamination in the sample cell, samples were loaded in the sample cell at 10 °C. Sample cell was thoroughly cleaned with ethanol and rinsed with DI water in between sample loadings. The collected data were baseline subtracted using the Origin software provided by the manufacturer.

The nanoemulsion solution was centrifuged using 4 mL Amicon ultra centrifugal filters with a 100 kDa cutoff. Centrifugation was performed at 4000 rpm for 30 min. To lower the viscosity, nanoemulsions were diluted fivefold prior to centrifugation. Subsequent to centrifugation, the supernatant (primarily composed of the droplet oil phase) was used to make a washed nanoemulsion with a final oil concentration of 0.4% wt. and Pluronic concentration of 1.2% wt. Subnatant solutions (continuous phase) were also collected and then mixed with Pluronic solution, with final concentration of 1.2% wt., and diluting the continuous phase fivefold with respect to our canonical concentration. The hydrodynamic diameter of the oil droplets remain unchanged after the centrifugation process.

**Shear rheology**. Rheological characterizations were conducted using a stress-controlled rheometer, Discovery Hybrid Rheometer, DHR3 (TA Instruments). An aluminium 60 mm 1.004° cone-plate fixture with a solvent trap geometry was utilized. The lower plate was equipped with a Peltier plate to control the temperature. The sol-to-gel transition was monitored in an oscillation temperature ramp, 2 °C/min, (from 10 °C to 50 °C) under strain amplitude of $\gamma_0 = 0.1\%$ and frequency of 20 rad/s. Prior to the temperature ramp, an oscillatory time sweep experiment for 300 s at 10 °C ($\gamma_0 = 0.1\%$ and $\omega = 20$ rad/s) was performed for temperature equilibration and to remove the shear history during the sample loading. After completion of gelation, all other rheological experiments were conducted. To capture large strain deformation behavior of the thermogelling nanoemulsion at 50 °C, large amplitude oscillatory shear (LAOS) tests were applied ($\gamma_0$ range of 0.08–50% and at $\omega = 20$ rad/s). To obtain frequency dependency of shear moduli in the thermogelling nanoemulsion (shown in Fig. 1b), the temperature ramp experiments were terminated at given temperatures to perform frequency sweep tests ($\gamma_0 = 0.1\%$). To capture flow and thixotropy behavior of the thermogelling nanoemulsion at 50 °C, flow sweep tests were applied from low to

high (0.1 1/s–100 1/s) and then high to low shear rate (100 1/s–0.1 1/s). For recovery behavior of thermogelling nanoemulsion at 50 °C, consecutive time sweep experiments were conducted at low (5 Pa for 300 s) and high (500 Pa for 120 s) oscillatory shear stress. In order to capture the speed of gelation from sol state to gel state of our thermogelling nanoemulsion, a temperature jump (20–50 °C) was applied while time sweep experiment ($\gamma_0 = 0.1\%$ and $\omega = 20$ rad/s) captured the shear moduli as a function of time.

## Data availability

The data that support the findings of this study are available from the corresponding author upon reasonable request.

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

## Acknowledgements
We acknowledge Dr. Samiul Amin for initiating this project and fruitful discussions. We acknowledge L'Oréal for financial support of the work. We also gratefully acknowledge Dr. Jan Ilavsky for his guidance on the small angle X-ray scattering results. This research used resources of the Advanced Photon Source, a U.S. Department of Energy (DOE) Office of Science User Facility operated for the DOE Office of Science by Argonne National Laboratory under Contract No. DE-AC02-06CH11357.

## Author contributions
S.M.H., A.Z.M.B., C.R.C., B.Z. and P.S.D. conceived the study. S.M.H and P.S.D. designed the experiments. S.M.H. conducted the experiments. S.M.H and P.S.D. wrote the paper and analyzed the data.

## Additional information

**Competing interests:** P.S.D. acknowledges financial support from L'Oréal. B.Z. is an employee of L'Oréal. The remaining authors declare no competing interests.

