## [Peer Review File · Nature Communications]

Reviewers' comments:

Reviewer #1 (Remarks to the Author):

The work is a very interesting paper describing the mechanisms occurring when a Pluronic copolymer system is added to a nanoemulsion.

I only have some minor questions and suggestions to the Authors.

In the Introduction the Authors suggest the possibility of using thermogelling formed hydrogels as vehicles for injectable product. In the case of the formulations described in the paper, have the Authors considered the feasibility of sterilization processes? Did they test the stability and the maintenance of thermogelling properties after steam sterilization? Or do they think about an alternative sterilization technique?

The Authors refer to nanoemulsions: indeed, emulsions whose mean diameters are lower than 150 nm are generally considered as microemulsions and they are thermodynamically stable and are transparent to visual observation. It is not clear the difference with the described nanoemulsions. Please, explain it in the text.

Mean diameters of nanoemulsions resulted in the 53-110 nm range, according to different modalities of cosurfactant addition. I think that the dilution of the nanoemulsion (required for DLS measurement) can significantly modify the droplet size, as a consequence of the reduction of both surfactant and cosurfactant concentration.

Similarly, the fivefold dilution of the nanoemulsion prior to centrifugation, can suggest modifications of droplet size. So, also micro-DSC measurements are performed in a system different from original nanoemulsion.

The rheological study is well conducted and exhaustive and the mechanisms involved in nanoemulsion gelification upon Pluronics addition are clearly explained. It would only be helpful to add a plot of viscosity trend of the described systems as a function of temperature, to underline the feasibility of injection.

As the Authors suggest the possibility to use the described systems in drug administration, did they try to dissolve API both in oil and/or aqueous phase?

Reviewer #2 (Remarks to the Author):

The authors report temperature-dependent rheological measurements of a gel system that contains microemulsion droplets, pluronic surfactants, and PEG. The authors' rheological characterizations are quite reasonable, yet others have previously made gel-like materials containing microemulsions of isopropyl myristate, a polar emollient, and polymer additives in a continuous aqueous phase. For instance, the authors' introduction and presentation do not mention the important and related prior work by Negi et al. (2016) in *Drug Delivery* 23 941 and by Carlfors et al. (1991) in *J. Dispersion Sci Tech.* 12 467. Both of these works, including the more recent Negi et al. publication, appropriately describe their systems involving isopropyl myristate with Tween type surfactants in a similar HLB range as microemulsions. The ingredients and approaches in these prior works are quite similar to the authors' so-called 'nanoemulsion made by low-energy emulsification'. So, the authors' 'nanoemulsion' component of their composition has already been presented as a microemulsion by others. Because these two prior works clearly and properly identify such systems as microemulsions, the authors' manuscript does not make appropriate contact with the existing related literature, which identifies such systems of nanoscale droplets, which have very similar compositions to the authors' composition, using the well-accepted and long-used definition, 'microemulsions'.

Could a high molecular weight non-polar oil that is very strictly insoluble in water be used in the same process as the authors' to make nanoscale droplets? The authors do not demonstrate this. The molecular weight of the one type of emollient (C17 'oil') used by the authors for the droplet material is relatively low. Moreover, this emollient is more polar than a simple n-alkane, so this will further enhance its quantitative solubility in polar water. It is well known that hexadecane is an oil that is not very strictly insoluble in water near standard conditions, even though it is immiscible with water, and hexadecane's residual solubility in water can lead to many effects, including

Ostwald ripening (see e.g. J. Weiss et al. 1999 Langmuir). Non-polar n-alkane oils having C-numbers around this range have been used in many kinds of microemulsion formulations for decades. So, the authors are controlling the rheology of a microemulsion system by using continuous-phase polymeric additives, just as Negi et al. enhanced their microemulsions with a continuous-phase hydrogel (that is known to have a temperature-dependent response) and obtained a soft-material gel that included microemulsion droplets.

While temperature-dependent gel-like materials can be made just of microemulsion phases themselves, the authors rely on adding pluronic block-copolymers to gel the system. The notion of gelling a system containing polymer components that associate differently at different temperatures has been established decades ago in the field of rheology, so the kind of temperature dependent behavior that the authors report is not surprising. The authors have not provided any direct imaging to substantiate their claim that a portion of the pluronic is actually in a large fraction of droplets; their experimental evidence for this hypothesized mechanism is indirect and therefore not very convincing.

Whether or not the polymer used by the authors remains entirely in the continuous phase or partially enters the droplets would not really matter very much if the polymer material is mostly in the continuous phase and is imparting the dominant rheological characteristics of the soft system. This is especially true at a relatively low droplet volume fraction of 0.24. Unless the rheological properties of the system could be controlled over a dramatically larger range (e.g. several orders of magnitude higher in modulus) at a volume fraction of 0.24 than at a volume fraction of e.g. 0.02 for identical polymer concentration in the continuous phase, temperature, etc., then there isn't much in this study that would be interesting from a rheological point of view. The absence of results related to droplet volume fraction dependence makes one think that the polymer additive(s) are effectively controlling the rheology of the system, and if that is the case, then the rheology of many pluronic solutions have already been well characterized as a function of concentration and temperature.

Other groups have clearly made water-based gel-like soft materials containing nanoscale droplets of isopropyl myristate with polymer additives (e.g. hydrogel formers), so this material concept has already been devised and published, even if there is a difference in the authors' choice of gelling additive. Many hydrogels have strongly temperature dependent rheological behavior, so this temperature-dependent aspect could be anticipated based on the prior use of hydrogels in very similar formulations. These prior materials used by other groups also are biocompatible, suitable for drug delivery or other applications. Thus, the novelty level and wow-factor do not meet high standard of Nature Communications. The authors' work, after appropriate revision to address the existing literature that captures the central idea in their introduction and to put their results in context of this prior literature, would be most appropriate in a field-specific journal, related to a field such as soft matter or rheology.

Reviewer #3 (Remarks to the Author):

The paper by Doyle et al. describes a thermoresponsive nanoemulsion gel formed by low-energy phase inversion method. Unlike previous work published by the group with high-energy methods, the gelator is a blend of FDA-approved Pluronic F127 and F68 triblock copolymers (PEO-PPO-PEO) in which the center of the copolymer is more hydrophobic. The authors propose a hydrophobic mechanism through which the Pluronic bridges the nanoemulsion droplets to form a colloidal gel. A mixture of three other cosurfactants (Tween, Span, PEG400) are also used in the stabilization of the nanoemulsion. DSC measurements are shown to support the argument that Pluronic binds to the surface of the bare nanoemulsions, along with standard rheological measurements of the viscoelastic moduli as a function of temperature. Overall, I am in favor of publication in NC because this organohydrogel system represents a significant step forward in bridging cosmetics and pharmaceutical products with academic model systems. Moreover, it is of interest to a broader material science/biomedical engineering community because of the material's stability, biocompatibility, and thermoresponsiveness. I have some major comments/questions and a few minor suggestions:

1) It is puzzling to me why a blend of F127 and F68 were used. It is also puzzling why so many other co-surfactants are used, as well as their roles in stabilizing the system. Theoretically speaking, only one Pluronic surfactant (the gelator) and one surfactant (Tween 80 or Span 20) should be necessary to stabilize the material. Why does PEG400 reduce the droplet size - wouldn't Tween and Span also perform the same function? Could the authors more carefully explain the role of each ingredient in the formulation?

1.1) Why is the PEG400 added at a different time during the nanoemulsion synthesis as compared to the rest of the ingredients?

2) Given the gelation mechanism proposed at moderate temperatures and the multiple previous studies in the group, it is important to show the microstructure of the gelled and ungelled systems using confocal microscopy or neutron scattering. While rheometry is important, and the data fits that of the Winter-Chambon criterion, it is a bulk technique that does not provide much insight into the microstructure. (As the saying goes: Rheology without morphology is theology.)

2.1) The authors state on p.3 that the DSC measurements support the gelation mechanism. DSC is designed to quantify the heats of reaction between molecules, but it does not indicate whether a sample is gelled at the mesoscale. Moreover, the "gel" temperatures measured by DSC differs from that from rheological measurements, and the temperature ramp rates are different on both instruments (2C/min vs 1C/min).

3) The gelation mechanism proposed by the authors could use more direct support from experimental data. First, even if the PPO block becomes more hydrophobic at higher temperatures and bind as in the schematic of Fig. 1, it is unclear to me how the PEO end groups induce bridging of the oil droplets. Are the PEO end groups charged? (If so, they might repel each other.) Could it be entanglement of the PEO chains? Or is it a depletion attraction introduced by the presence of high volume fraction of PEG400/Pluronic micelles in the suspension phase? These are all plausible mechanisms that the authors should systematically discuss. The authors speculate that hydrogen bonding is a possible mechanism without additional elaboration (lines 262-263).

4) On line 190, the authors state that the jamming of oil droplets lead to thermogelation in the system. This could use further support, given that the effective oil volume fraction is estimated to be between 24% to 43%, far below the random closed packing fraction of hard spheres (64%). In pure Pluronic systems, the molecules form crystals and other structures at high volume fractions [ref]. They do not form gels. Furthermore, this statement seems to contradict the authors' discussion of the mechanism in the text and in Fig. 1. Please explain.

Other comments:

1) Fig. S3 shows important control data but is a bit messy. It should include data for Pluronic concentration between 4.6% and 13%. The rheology for various concentrations of PEG400 and the Tween/Span mixtures (holding one fixed while varying the other) should be systematically measured.

1.1) From Fig. S3 c-e, it appears as though the effect of Pluronic on the gelling system is minimal.

2) In purifying the nanoemulsions, does centrifuging the material at 4000 rpm at 30 minutes not aggregate or deform the oil droplets? Has particle sizing been performed before and after centrifuging to verify droplet stability?

3) In Fig. 6b, the gelled nanoemulsions form interesting shapes when dropped into a heated water bath. What is the persistence time of these shapes before they dissolve in the water?

4) The viscosity of the gelled samples above the gel point could be provided in Fig. 6a as a comparison against the ungelled nanoemulsions.

5) At what shear rates were the data in Fig. S3b collected? The gel points here are a little different

from the ones shown in the main manuscript.

We would like to thank the reviewers for their insightful and constructive comments. Below are our responses to their specific comments.

A summary of the major changes to manuscript are as follows:

- We highlighted in yellow all the added content/revisions in the main text and SI files and elaborate on these below.
- We added 5 more references (# 37, 38, 39, 57, and 63) into the main text.
- We added 7 more figures with new data in the SI file (Fig. S4, Fig. S5, Fig. S6, Fig. S8, Fig. S9, Fig. S10, and Fig. S11) and we also updated our previous Fig. S1 and Fig. S2 and added Fig. S1 b, c and Fig. S2a.
- We have added a discussion for each image in the SI file.

Reviewers' comments and replies:

Reviewer #1 (Remarks to the Author):

The work is a very interesting paper describing the mechanisms occurring when a Pluronic copolymer system is added to a nanoemulsion.

I only have some minor questions and suggestions to the Authors.

In the Introduction the Authors suggest the possibility of using thermogelling formed hydrogels as vehicles for injectable product. In the case of the formulations described in the paper, have the Authors considered the feasibility of sterilization processes? Did they test the stability and the maintenance of thermogelling properties after steam sterilization? Or do they think about an alternative sterilization technique?

The formed thermogelling nanoemulsion requires further exploration for transdermal and topical formulation application. In terms of sterilization processes, heat sterilization or sterile filtration techniques can be performed. Although heat sterilization may provide a better sterilization, it is not recommended as exposing the nanoemulsion to the steam temperature may accelerate Ostwald ripening, specifically for our metastable nanoemulsion with nonionic emulsifiers. However, one can perform the sterilization by application of gamma-rays or UV light which should not disrupt the nanoemulsion. In addition to the above strategies, we could sterilize all the ingredients and tools prior to the formation of the nanoemulsion (under sterile condition).

The Authors refer to nanoemulsions: indeed, emulsions whose mean diameters are lower than 150 nm are generally considered as microemulsions and they are thermodynamically stable and are transparent to visual observation. It is not clear the difference with the described nanoemulsions. Please, explain it in the text.

The reviewer brings up a subtle point which has led to some confusion in the literature. As stated correctly by the reviewer, microemulsions are thermodynamically stable dispersions and are typically very small (<150 nm or so). In contrast, nanoemulsions are kinetically stable emulsion droplets (not thermodynamically stable) whose size is generally smaller than ~500nm (the exact cutoff in size from a macroemulsions is not well established). There is a nice review of these subtle differences in "Nanoemulsions versus microemulsions: terminology, differences and similarities" by McClements (Soft Matter, 8, 1719, 2012). To clarify this point, the main text was revised (highlighted text in page 2 and page 3) to clearly mention this difference and we also added a discussion in the SI file (Fig S2a) to discern nanoemulsion and versus microemulsion formation and verify that our system is indeed a nanoemulsion.

Mean diameters of nanoemulsions resulted in the 53-110 nm range, according to different modalities of cosurfactant addition. I think that the dilution of the nanoemulsion (required for DLS measurement) can significantly modify the droplet size, as a consequence of the reduction of both surfactant and cosurfactant concentration.

Nanoemulsions are not as sensitive as microemulsions to changes in composition (e.g. dilution or addition of other molecules) *after* their formation – nanoemulsions are metastable systems. This metastability allowed us to form different nano-droplet sizes (e.g. as shown in Fig. 4), yet having the exact same final composition. Similarly, the dilution of nanoemulsion for DLS measurement does not affect the droplet size (at least during the course of our experimental measurements). In fact, we showed that droplet size remained almost unchanged after several hours after being diluted 200 fold. To clarify this point, we added this information in the SI (new Fig. S2a) and main text (highlighted text in page 3).

Similarly, the fivefold dilution of the nanoemulsion prior to centrifugation, can suggest modifications of droplet size.

The hydrodynamic diameter of the nanoemulsions was measured and monitored during all steps involved in the centrifugation process. The DLS measurements showed that the oil droplet size remained constant during the dilution and centrifugation processes. We added a sentence in the text to clarify this matter (Page 14, highlighted text).

The rheological study is well conducted and exhaustive and the mechanisms involved in nanoemulsion gelification upon Pluronic addition are clearly explained. It would only be helpful to add a plot of viscosity trend of the described systems as a function of temperature, to underline the feasibility of injection.

We would like to thank the reviewer for this useful suggestion and comments on the rheological study of our system. We performed viscosity measurements as a function of temperature and added these new results in the SI file (Fig. S8) along with a discussion (Page 9 in the SI).

As the Authors suggest the possibility to use the described systems in drug administration, did they try to dissolve API both in oil and/or aqueous phase?

Indeed, we did examine the release of an active pharmaceutical ingredient (API) in the sol and gel state. A new figure was added in the SI summarizing the results from these experiments (Fig. S9). A poorly water-soluble model API, ibuprofen, was dissolved in the oil phase and its dissolution monitored below and above gelation point. A short discussion was added in the main text (highlighted text in page 11 and 12). We thank the reviewer for suggesting that we add in these new results.

Reviewer #2 (Remarks to the Author):

The authors report temperature-dependent rheological measurements of a gel system that contains microemulsion droplets, pluronic surfactants, and PEG. The authors' rheological characterizations are quite reasonable, yet others have previously made gel-like materials containing microemulsions of isopropyl myristate, a polar emollient, and polymer additives in a continuous aqueous phase. For instance, the authors' introduction and presentation do not mention the important and related prior work by Negi et al. (2016) in *Drug Delivery* 23 941 and by Carlfors et al. (1991) in *J. Dispersion Sci Tech.* 12 467. Both of these works, including the more recent Negi et al. publication, appropriately describe their systems involving isopropyl myristate with Tween type surfactants in a similar HLB range as microemulsions. The ingredients and approaches in these prior works are quite similar to the authors' so-called 'nanoemulsion made by low-energy emulsification'. So, the authors' 'nanoemulsion' component of their composition has already been presented as a microemulsion by others. Because these two prior works clearly and properly identify such systems as microemulsions, the authors' manuscript does not make appropriate contact with the existing related literature, which identifies such systems of nanoscale droplets, which have very similar compositions to the authors' composition, using the well-accepted and long-used definition, 'microemulsions'.

We appreciate the reviewer comments on previous works by Negi et al. (2016) and Carlfors et al. (1991). We added these papers in the reference list (refs. # 37 and 39).

First, we would like to reiterate that there is confusion in the literature regarding a nanoemulsion versus a microemulsion – these are not the same. Please see above our reply to reviewer #1 on this point and the added discussion in the SI regarding our system and all the facts supporting that it is a nanoemulsion (metastable dispersion) and not a microemulsion (thermodynamic equilibrium state).

Negi et al. (2016) used Carbopol 934 to gel their phospholipid-based microemulsion and investigated the permeability of active drug molecules in comparison to the commercial cream. They used saturated lipids and Tween 80 to emulsify the oil phase with ethanol as a cosurfactant. Firstly, they formed a microemulsion which is a thermodynamic stable system (as they also defined it in the introduction section). As we mentioned in the main text, one of the disadvantages of thermodynamically stable system is that they are very sensitive to thermodynamic parameters variation such as concentration and temperature. In contrast, nanoemulsions are desirable in applications as the formulation remains comparably intact during processing and transportation. Secondly, Negi et al. (2016) used Carbopol (which is a typical gelling material or thickening agent) to obtain a gel like emulsion system. As we mentioned in the introduction one strategy to obtain a gel in emulsion based system, is using a component to gel the continuous phase. In these systems if the oil droplets are removed from the formulation, the gel formation still occurs in the continuous phase. However, in our presented thermogelling nanoemulsion oil droplets play a key synergistic role in the observed gel formation – there is no gel if our nanoemulsion is removed from our formulation.

Carlfors et al. (1991) fabricated what they believed to be a thermodynamically stable oil-in-water as a topical delivery material. Their system has no thermally-gelling response. In fact, they report that their system becomes bicontinuous at elevated temperatures – we see no such transition in our system (see our added SAXS data in the SI, for example). There is no rheology reported in this article, most probably because there is nothing remarkable which

occurs in the rheology. They assumed that the system was a microemulsion if the system was clear, homogenous and qualitatively low-viscosity (apparently by the eye). They report that some systems were checked to be in equilibrium by centrifugation, though it is not clear which and how many. In contrast, we show in our work that the order of mixing components (having the same final composition) gives rise to 2 different sets of metastable nanoemulsions with differing diameters (see new SI section "Optimizing the oil droplet size in the nanoemulsion") - these are not equilibrium systems. Further discussion and data supporting that our system is a nanoemulsion is given in the new SI section "Verification of the formed nanoemulsion and its stability".

Could a high molecular weight non-polar oil that is very strictly insoluble in water be used in the same process as the authors' to make nanoscale droplets? The authors do not demonstrate this. The molecular weight of the one type of emollient (C17 'oil') used by the authors for the droplet material is relatively low. Moreover, this emollient is more polar than a simple n-alkane, so this will further enhance its quantitative solubility in polar water. It is well known that hexadecane is an oil that is not very strictly insoluble in water near standard conditions, even though it is immiscible with water, and hexadecane's residual solubility in water can lead to many effects, including Ostwald ripening (see e.g. J. Weiss et al. 1999 Langmuir). Non-polar n-alkane oils having C-numbers around this range have been used in many kinds of microemulsion formulations for decades. So, the authors are controlling the rheology of a microemulsion system by using continuous-phase polymeric additives, just as Negi et al. enhanced their microemulsions with a continuous-phase hydrogel (that is known to have a temperature-dependent response) and obtained a soft-material gel that included microemulsion droplets.

We respectfully disagree with the reviewer's comment on controlling the rheology of our nanoemulsion by simply adding nanoemulsion to a continuous phase which would otherwise gel on its own (as was done by Negi et al.). Also, the rheological properties in the presented system can be controlled by multiple parameters, including oil fraction (Fig. S10), oil droplet size (Figure 4), Pluronic concentration (Fig. S11), Pluronic type (Fig. S7), and PEG concentration (Figure 5) – these are all supportive of the synergistic gelation mechanism we proposed.

We have used different oils to prepare nanoemulsions, including oleic acid, squalene, ethyl laurate, and caprylic capric (composition is similar to our canonical formulation). However, in most cases the oil droplets size obtained via the PIC emulsification process resulted in a relatively large droplet diameter. For example, a nanoemulsion using squalene oil (chemical formula of squalene is $C_{30}H_{50}$, considered to be very hydrophobic oil) resulted in oil droplet size ~ 260 nm (PDI: 0.35) without a strong thermogelling behavior. However, using ethyl laurate as an oil phase resulted in a hydrodynamic oil droplet of ~ 60 nm, which forms a gel at an elevated temperature. As we also indicated in the manuscript (Figure 4), we speculate that the thermogelling behavior is highly correlated with the oil droplet size.

While temperature-dependent gel-like materials can be made just of microemulsion phases themselves, the authors rely on adding pluronic block-copolymers to gel the system. The notion of gelling a system containing polymer components that associate differently at different temperatures has been established decades ago in the field of rheology, so the kind of temperature dependent behavior that the authors report is not surprising. The authors have not provided any direct imaging to substantiate their claim that a portion of the pluronic is actually in a large fraction of droplets; their experimental evidence for this hypothesized mechanism is indirect and therefore not very convincing.

The reviewer does not fully appreciate a key point of the manuscript- gelation in our system does not occur solely due to the added Pluronics, nor with only the nanoemulsions – it is the synergistic effect of the two components which gives rise to the rich physics seen. This is new, not intuitive, and has not been established in the literature.

Regarding direct imaging of the small oligomers onto the surface of the oil droplet – this is not feasible in our system. The scale of this problem, namely a ~50 nm liquid droplet interface combined with a small oligomer, makes it unfeasible to directly image the process. We do not have knowledge of a technique which would allow for imaging at this length scale and on liquid interfaces.

That said, we did provide quantitative and carefully constructed micro-DSC experiments to validate the adsorption of the copolymers onto the oil droplet interfaces. The micro-DSC captures the endo- or exo-thermic interactions between components in the solution as a function of temperature. This is a well-established technique used to study molecular-level adsorption and association processes. For example, other researchers investigated the interaction between Pluronic and nonionic surfactants using micro-DSC analysis (DOI:10.1021/la0104267), or interaction between Pluronic and ionic surfactants (DOI: 10.1021/jp012729f), or effect of salt on aqueous solution properties of Pluronic (DOI: 10.1021/la9703712).

In addition to the micro-DSC data, our hypothesis is supported by the rheology experiments (a new discussion was added for the Fig. S3). The concentration of the copolymer used in this work is much lower than that to just simply gel the continuous phase. We demonstrate this in our control rheology experiments (Fig. S3).

The reviewer's comment did make us reflect on what direct imaging we could carry out. Confocal microscopy (which is the only practical imaging technique for our system) was utilized to image the microstructure of our nanoemulsion. The newly obtained results were added in the SI file (Fig. S4) and main text (highlighted text in page 6).

Whether or not the polymer used by the authors remains entirely in the continuous phase or partially enters the droplets would not really matter very much if the polymer material is mostly in the continuous phase and is imparting the dominant rheological characteristics of the soft system. This is especially true at a relatively low droplet volume fraction of 0.24. Unless the rheological properties of the system could be controlled over a dramatically larger range (e.g. several orders of magnitude higher in modulus) at a volume fraction of 0.24 than at a volume fraction of e.g. 0.02 for identical polymer concentration in the continuous phase, temperature, etc., then there isn't much in this study that would be interesting from a rheological point of view. The absence of results related to droplet volume fraction dependence makes one think that the polymer additive(s) are effectively controlling the rheology of the system, and if that is the case, then the rheology of many pluronic solutions have already been well characterized as a function of concentration and temperature.

The reviewer is correct in noting that it is important to show that the oil phase plays a critical role in determining the rheological response in the system and that it is not just the polymer controlling the rheology. Indeed, we did show this in our control experiments in which we removed the oil and only looked at the rheology of the remaining polymers – this data is shown in the SI Fig. S3. Comparing this data to the main text Fig. 2, we note that indeed the elastic modulus changes by > 4 orders of magnitude by adding a small amount (volume fraction of 0.24) of oil droplets to the system. This data not only supports our mechanism, but clearly differentiates our work from the prior literature in which the gelling continuous phase was agnostic to the presence of the oil nanoemulsions.

We thank the reviewer for the suggestion of varying the oil fraction. We performed new experiments and added the results in Fig. S10 (different oil fraction). We also varied the gelator concentration and the new results are shown in Fig. S11. For example, G' is ~ 4,000 Pa at an oil fraction of 0.18, and increases to 10,000 Pa when oil fraction is ~ 0.35.

Other groups have clearly made water-based gel-like soft materials containing nanoscale droplets of isopropyl myristate with polymer additives (e.g. hydrogel formers), so this material concept has already been devised and published, even if there is a difference in the authors' choice of gelling additive. Many hydrogels have strongly temperature dependent rheological behavior, so this temperature-dependent aspect could be anticipated based on the prior use of hydrogels in very similar formulations. These prior materials used by other groups also are biocompatible, suitable for drug delivery or other applications. Thus, the novelty level and wow-factor do not meet high standard of Nature Communications. The authors' work, after appropriate revision to address the existing literature that captures the central idea in their introduction and to put their results in context of this prior literature, would be most appropriate in a field-specific journal, related to a field such as soft matter or rheology.

We respectfully disagree with the main point above, namely we know of no other prior studies which show thermal gelation of nanoemulsions through the *synergistic* effects described in our article. The reviewer is correct that others have put nanoemulsions into continuous phases to form thermally gelling formulations, however in the prior work the continuous phase would have thermally gelled in of itself regardless of the nanoemulsion – our system will not. We have added one more reference in the article to clarify this point (ref. # 37). New physics were discovered in our work showing a subtle synergy of the two components (Pluronics and nanoemulsions) and a system developed which has potential applications due the choice of FDA approved components and a low energy formation method.

Reviewer #3 (Remarks to the Author):

The paper by Doyle et al. describes a thermoresponsive nanoemulsion gel formed by low-energy phase inversion method. Unlike previous work published by the group with high-energy methods, the gelators is a blend of FDA-approved Pluronic F127 and F68 triblock copolymers (PEO-PPO-PEO) in which the center of the copolymer is more hydrophobic. The authors propose a hydrophobic mechanism through which the Pluronic bridges the nanoemulsion droplets to form a colloidal gel. A mixture of three other cosurfactants (Tween, Span, PEG400) are also used in the stabilization of the nanoemulsion. DSC measurements are shown to support the argument that Pluronic binds to the surface of the bare nanoemulsions, along with standard rheological measurements of the viscoelastic moduli as a function of temperature. Overall, I am in favor of publication in NC because this organohydrogel system represents a significant step forward in bridging cosmetics and pharmaceutical products with academic model systems. Moreover,

it is of interest to a broader material science/biomedical engineering community because of the material's stability, biocompatibility, and thermoresponsiveness. I have some major comments/questions and a few minor suggestions:

We thank the reviewer for the positive comments and feedback.

1) It is puzzling to me why a blend of F127 and F68 were used. It is also puzzling why so many other co-surfactants are used, as well as their roles in stabilizing the system. Theoretically speaking, only one Pluronic surfactant (the gelator) and one surfactant (Tween 80 or Span 20) should be necessary to stabilize the material. Why does PEG400 reduce the droplet size - wouldn't Tween and Span also perform the same function? Could the authors more carefully explain the role of each ingredient in the formulation?

Nanoemulsion formation through a low energy method greatly depends on the hydrophilic lipophilic balance (HLB) of the emulsifiers (see for example our prior article Gupta et al. Langmuir 2017). In order to optimize the oil droplet size, we varied the HLB of the emulsifiers (hence a use of both Tween and Span). Instead of using a different surfactant with different HLB, we used two emulsifiers, with very different HLB values, and parametrically varied them to minimize the oil droplet size. Tween 80 is more hydrophilic emulsifier (HLB ~ 15) and Span 80 has more lipophilic characteristic (HLB ~ 4.3). This help us to leverage HLB values during emulsification without significantly changing the chemistry of the emulsifiers. To further reduce the oil droplet size, PEG molecules were used as a cosurfactant. We added a section in the SI Fig S1b and Fig. S1c and a discussion on optimizing the oil droplet size to discuss this issue in more detail.

We also added a relatively small amount of F68 in addition to F127 (as the gelator mixture) to mainly manipulate the viscoelastic behavior and gel point of the system. For example, if only F127 is used as the gelator (with similar molar fraction), G' is about one order of magnitude higher that of mixture of gelators (F127 / F68 : 6/1). Addition of F68 with higher LCST would increase the flow behavior of the nanoemulsion suspension at ambient temperature. We added this point in the main text (page 13 in 'Synthesis of thermogelling nanoemulsion' section) to explain why a small amount of F68 was used.

1.1) Why is the PEG400 added at a different time during the nanoemulsion sythesis as compared to the rest of the ingredients?

We used PEG 400 during the emulsification process since it acts as a cosurfactant. However, in Figure 4 and 5 we added part or all of the PEG 400 after nanoemulsion formation in order to have a similar composition while varying oil droplet size (Figure 4) or fixing a constant oil droplet size while increasing PEG concentration (Figure 5). We added a sentence in the method section to more clearly point this out (Page 13, highlighted text in the second paragraph).

2) Given the gelation mechanism proposed at moderate temperatures and the multiple previous studies in the group, it is important to show the microstructure of the gelled and ungelled systems using confocal microscopy or neutron scattering. While rheometry is important, and the data fits that of the Winter-Chambon criterion, it is a bulk technique that does not provide much insight into the microstructure. (As the saying goes: Rheology without morphology is theology.)

We are not lined up at this point in time to perform neutron scattering. We did however perform confocal microscopy. We added confocal microscopy data (Fig. S4) along with a discussion in the main text (highlighted text in page 6). We note that the gelling mechanism here is different than our prior work using PEGDA wherein the latter resulted in large micron-sized domains due to arrested phase separation. In the current work, we expect the structure has length scales comparable to the emulsion droplet size and less than the wavelength of visible light. This is consistent with what we observe in confocal experiments. We also performed x-ray scattering on one system (Fig. S5). The SAXS confirms that the droplets remain intact at elevated temperatures and do not form droplet clusters, which is consistent with the confocal imaging and our proposed mechanism.

2.1) The authors state on p.3 that the DSC measurements support the gelation mechanism. DSC is designed to quantify the heats of reaction between molecules, but it does not indicate whether a sample is gelled at the mesoscale. Moreover, the "gel" temperatures measured by DSC differs from that from rheological measurements, and the temperature ramp rates are different on both instruments (2C/min vs 1C/min).

We agree with the reviewer that DSC measurements cannot be used directly to find if the sample forms a gel. However, DSC data support the interaction between oil and the copolymers as temperature increases. This adsorption is believed to play a major role in the observed thermogelling behavior of our nanoemulsion.

We thank the reviewer for pointing out the differences in temperature ramping rate in the micro-DSC and rheology. Although the temperature ramping in the micro-DSC seems slower than rheology (1 °C/min vs 2 °C/min), but the

overall measurement time is similar. We added information regarding this on page 14, in the DSC section. However, the direct comparison regarding the gelation temperature between these measurements may not be feasible since the micro-DSC analyses were performed at a diluted state due to limits of the measurement apparatus.

3) The gelation mechanism proposed by the authors could use more direct support from experimental data. First, even if the PPO block becomes more hydrophobic at higher temperatures and bind as in the schematic of Fig. 1, it is unclear to me how the PEO end groups induce bridging of the oil droplets. Are the PEO end groups charged? (If so, they might repel each other.) Could it be entanglement of the PEO chains? Or is it a depletion attraction introduced by the presence of high volume fraction of PEG400/Pluronic micelles in the suspension phase? These are all plausible mechanisms that the authors should systematically discuss. The authors speculate that hydrogen bonding is a possible mechanism without additional elaboration (lines 262-263).

We thank the reviewer for raising these important points regarding the gelation mechanism of our system.

We have not changed the chemistry of the Pluronic copolymers and they are used as received. Thus the PEG blocks are nonionic. The micro-DSC experiments (Figure 3) were conducted at a much lower concentration of the canonical system (presented in Figure 2), and therefore we were unable to observe a possible colloidal crystal peak (a secondary endothermic peak) as it was previously reported for concentrated Pluronic solution (e.g. Ref. #51).

We do not believe that the gelation is due to depletion attraction. The F127 micelles are rather large (~20nm) and so from a colloidal point of view this system at high temperature is more akin to a bi-disperse suspension. Also, if the gelation was due to depletion, we would expect the doubling of the PEG concentration in Figure 5 to have a more dramatic effect on the plateau gel modulus. Lastly, depletion gels tend to form large clusters which we did not observe in confocal microscopy (new confocal data was added to SI).

It is well established that micellization of Pluronics at elevated temperatures is due to the dehydrated PPO which form the core of the micelles. Our micro-DSC measurements (Figure 3) strongly suggest that the PPO adsorption on the oil interface is in fact more favorable than micellization. This result lends strong support to one key part of our proposed mechanism, namely that the nanoemulsions become decorated with the Pluronic to become “hairy spheres” at elevated temperatures. In fact, this decoration of the nanoemulsion surface most likely precedes the formation of Pluronic micelles (see Figure 3) and may explain why our temperature sweep data (e.g. G' versus T in Fig. 2) do not show as abrupt a transition as seen in the more concentrated *pure* Pluronic solutions (Fig. S3). Furthermore, prior work has shown that Pluronics form polymer brushes when adsorbed onto hydrophobic surfaces (see for example this work with F127 DOI: 10.1021/la9001169). As described below, overlap of the PEO brushes is what is thought to give rise to the elasticity of pure Pluronic systems at elevated temperatures.

Thus far we have established that nanoemulsions become decorated with the Pluronic at elevated temperatures. It is also well-accepted that the Pluronics themselves will form micelles at increased temperature – thus in our system at elevated temperatures we have a collection of hairy nanoemulsions (diameters ~50nm) with comparably “hairy” micelles (PPO core and the solvated PEO chains, diameters ~20nm). To understand the origin of the elasticity of our system, we must first recall what happens during gelation in the pure Pluronic systems. Seminal work by Prud'homme and coworkers (Langmuir 1996), showed that at elevated temperature the micelles increase in number to eventually form an ordered phase (cubic packing) and the physical source of the elasticity (and yield stress) in this system arises from the overlapping micelle shells formed of PEO chains. There have been several studies since the work of Prud'homme et al. which support and have expanded on this basic picture of the source of elasticity in the Pluronic systems. For example, more recent work by Lau et al. (J. Polymer Sci. B 2004), suggest that the PEO chains serve as transient “physical junctions” in a network. They also suggested these transient junctions are related to the hysteresis in G' observed when heating up and then cooling a system.

Our system is obviously more complicated than the pure Pluronics, but at a coarse-grained scale it shares similarities – concentrated packings of hairy colloidal objects composed of nanoemulsions and micelles. In analogy with the pure Pluronic copolymer solution, we speculate that the origin of the elasticity and gel-like response in our thermogelling nanoemulsion is most likely attributed to a sequence of events as temperature is increased: 1) adsorption of the Pluronic to the nanoemulsion, 2) Pluronic micelle formation, and 3) increase in Pluronic micelles until they become close-packed such that there will be overlap of PEO chains amongst the various entities (both micelles *and* hairy droplets), as depicted in Figure 1c. This is more evident in a temperature ramping cycle from heating to cooling and then reverse. We added Fig. S6 and a paragraph in the SI as well as a discussion part in the main text (page 7) to clarify this point. Currently we are unsure about the exact structure formed (i.e. is there a cubic lattice or something else) and future studies will be performed, e.g. by neutron scattering techniques, to further investigate the structural evolution in our system.

4) On line 190, the authors state that the jamming of oil droplets lead to thermogelation in the system. This could use further support, given that the effective oil volume fraction is estimated to be between 24% to 43%, far below the random closed packing fraction of hard spheres (64%). In pure Pluronic systems, the molecules form crystals and other structures at high volume fractions [ref]. They do not form gels. Furthermore, this statement seems to contradict the authors' discussion of the mechanism in the text and in Fig. 1.

As the reviewer mentioned, 0.64 is considered as the critical volume fraction of the random closed packing of hard sphere system. As we mentioned, the adsorption of the copolymer to the oil interfaces results in the increasing the effective volume fraction of the oil droplets. For example, in our canonical formulation we estimated the growth of the effective volume fraction to be as high as 0.43. In addition, an increased concentration of mixed micelles formation is expected in the continuous phase (also depicted in Figure 1c). The mixed micelle formation was previously reported from DSC data (Ref. # 56, Langmuir 2001, 17, 4818-4824). Therefore, combination of the formed hairy oil droplets and micelles formation leads to the packed structure sketched in Figure 1c, it is not just the droplets themselves.

Other comments:

1) Fig. S3 shows important control data but is a bit messy. It should include data for Pluronic concentration between 4.6% and 13%. The rheology for various concentrations of PEG400 and the Tween/Span mixtures (holding one fixed while varying the other) should be systematically measured.

The aim of the control rheology measurements, Fig. S3, was to show first, the minimum Pluronic concentration to obtain a gel (we found it is greater than 15% wt.) at an elevated temperature. Secondly, no gel formation is obtained in the continuous phase of our canonical nanoemulsion formulation (Fig. S3 c, d, and e). We added a discussion part in the SI for the Fig. S3 to clarify the control rheology experiments in detail.

We added Fig. S9 and Fig. S10 where we systematically vary the oil fraction and Pluronic concentration in our system along with a discussion (page 10 in SI file).

1.1) From Fig. S3 c-e, it appears as though the effect of Pluronic on the gelling system is minimal.

Fig. S3 c-e have been designed not to change the Pluronic concentration, but considering different scenarios in terms of the aqueous phase composition. We added a detailed discussion in the SI to thoroughly discuss these points.

2) In purifying the nanoemulsions, does centrifuging the material at 4000 rpm at 30 minutes not aggregate or deform the oil droplets? Has particle sizing been performed before and after centrifuging to verify droplet stability?

Thank you for mentioning this point. Hydrodynamic diameter was monitored during the steps involved in the micro-DSC experiments. We did not notice any size change after centrifugation or dilution. We added this point in page 14.

3) In Fig. 6b, the gelled nanoemulsions form interesting shapes when dropped into a heated water bath. What is the persistence time of these shapes before they dissolve in the water?

The gelled nanoemulsions shown in Figure 6c and video in the SI will eventually fully dispersed (in over ~ 1 hr) in the excess water (total volume of water in the beaker was ~ 100 mL). We modified the wording in the first line of page 9 in the manuscript to clarify this point.

4) The viscosity of the gelled samples above the gel point could be provided in Fig. 6a as a comparison against the ungelled nanoemulsions.

We added Fig. S8 with a small paragraph (page 9) in SI.

5) At what shear rates were the data in Fig. S3b collected? The gel points here are a little different from the ones shown in the main manuscript.

Fig. S3b was conducted at a constant shear stress of 0.1 Pa. We have revised the caption to contain this information.

Fig. S3b displays the evolution of G' and G'' as a function of temperature for pure Pluronic solution, whereas in the main text we have not shown any Figures for pure Pluronic. All Figures shown in the main text are for thermogelling nanoemulsions.

REVIEWERS' COMMENTS:

Reviewer #1 (Remarks to the Author):

The manuscript has been revised according to my previous suggestions and observations and I have no further observations to do

Reviewer #2 (Remarks to the Author):

The authors report temperature-dependent rheological measurements of a gel system that contains microemulsion droplets, pluronic surfactants, and PEG. The authors' rheological characterizations are quite reasonable, yet others have previously made gel-like materials containing microemulsions of isopropyl myristate, a polar emollient, and polymer additives in a continuous aqueous phase. For instance, the authors' introduction and presentation do not mention the important and related prior work by Negi et al. (2016) in *Drug Delivery* 23 941 and by Carlfors et al. (1991) in *J. Dispersion Sci Tech.* 12 467. Both of these works, including the more recent Negi et al. publication, appropriately describe their systems involving isopropyl myristate with Tween type surfactants in a similar HLB range as microemulsions. The ingredients and approaches in these prior works are quite similar to the authors' so-called 'nanoemulsion made by low-energy emulsification'. So, the authors' 'nanoemulsion' component of their composition has already been presented as a microemulsion by others. Because these two prior works clearly and properly identify such systems as microemulsions, the authors' manuscript does not make appropriate contact with the existing related literature, which identifies such systems of nanoscale droplets, which have very similar compositions to the authors' composition, using the well-accepted and long-used definition, 'microemulsions'.

We appreciate the reviewer comments on previous works by Negi et al. (2016) and Carlfors et al. (1991). We added these papers in the reference list (refs. # 37 and 39).

> It is good to add these works in the list of references, and it would be most beneficial to cite these in the introductory part of the main text as yielding droplets in the same size range with a similar composition and approach to formation.

First, we would like to reiterate that there is confusion in the literature regarding a nanoemulsion versus a microemulsion – these are not the same. Please see above our reply to reviewer #1 on this point and the added discussion in the SI regarding our system and all the facts supporting that it is a nanoemulsion (metastable dispersion) and not a microemulsion (thermodynamic equilibrium state).

> This confusion in terminology has already been identified in past articles on the subject, of which this reviewer has been aware throughout. The terminology associated with nanoemulsions is in fact tied to the route of formation in addition to the metastability of the system produced. In fact, the confusion in the literature lies in the fact that microemulsion formation is not nanoemulsion formation by a low-energy process, since nanoemulsion formation requires significant external non-thermodynamic driving forces to create the droplets. McClements review is not by any means authoritative over other earlier publications on the subject that had already differentiated between microemulsions and nanoemulsions in this manner. Diluted microemulsions have been studied for decades by small angle x-ray scattering and small angle neutron scattering. In such published studies, these systems are still referred to as microemulsions, even if they have been diluted. That is because the route of forming such small droplets is making a microemulsion through thermodynamic means (i.e. usually involving a drastic reduction in interfacial tension between two immiscible liquid phases) not through imposing external flow that induces droplet rupturing. It is well known that the microemulsion droplet structures, once formed, can often, although not always, be preserved over significant times by diluting. In the literature, there is precedent for still calling such diluted systems of nanoscale droplets formed in this way as 'microemulsions'. By contrast, as first defined, nanoemulsions are formed by a flow-rupturing process, not by a thermodynamic microemulsion process and subsequent dilution. The author's droplet system would most appropriately be described as a 'diluted microemulsion'. This would provide the reader

with a better view of the method of formation. Even if the authors claim that dilution causes their droplets to enter a region of the phase diagram for which droplets are not stable (i.e. diluting to produce metastability), their route of obtaining droplets still involves microemulsion production, and the reader needs to be informed of this directly. If the authors do not provide a long-duration study of an equilibrium phase diagram of the specific composition involved, it is difficult to determine if the final diluted compositions of the authors is even in a metastable region. If it is metastable, then the authors need to provide bounds on the coarsening rate of the droplet sizes (see Reviewer #1's comment related to this) or what they type of structure, if different than droplets, would be.

Negi et al. (2016) used Carbopol 934 to gel their phospholipid-based microemulsion and investigated the permeability of active drug molecules in comparison to the commercial cream. They used saturated lipids and Tween 80 to emulsify the oil phase with ethanol as a cosurfactant. Firstly, they formed a microemulsion which is a thermodynamic stable system (as they also defined it in the introduction section). As we mentioned in the main text, one of the disadvantages of thermodynamically stable system is that they are very sensitive to thermodynamic parameters variation such as concentration and temperature. In contrast, nanoemulsions are desirable in applications as the formulation remains comparably intact during processing and transportation. Secondly, Negi et al. (2016) used Carbopol (which is a typical gelling material or thickening agent) to obtain a gel like emulsion system. As we mentioned in the introduction one strategy to obtain a gel in emulsion based system, is using a component to gel the continuous phase. In these systems if the oil droplets are removed from the formulation, the gel formation still occurs in the continuous phase. However, in our presented thermogelling nanoemulsion oil droplets play a key synergistic role in the observed gel formation – there is no gel if our nanoemulsion is removed from our formulation.

> The main point here is that the concept of microemulsion droplets in a water-based polymeric matrix (i.e. hydrogel) with controllable soft-elastic rheology has already been introduced and is in the literature through Negi et al (2016). So the concept and first demonstration of a hydrogel-microemulsion (i.e. nanoscale droplet) delivery system is in the literature already, even if the specific characteristics and composition of the authors' system is different. Readers should know this up front. Relabeling microemulsion to nanoemulsion would not change this point, since the droplet formation mechanism is essentially the same in Negi et al. even if the type of other polymeric additive is different. The point that the authors make-- that there is no gel if the droplets are removed from their composition-- is not lost on the reviewer. Synergistic gelation effects are known in rheology for a lot of different systems -- even simpler compositions of dissolved salts in an aqueous solution and a dispersion of droplets reasonably far below the jamming point, both of which would be simply viscous if measured separately. While there appear to be no prior publications on the exact composition that the authors have used, the concept of adding one water-based material to another and finding gelation isn't new to solution-colloid, polymer-colloid, or colloid-colloid mixtures. Often colloidal self-assembly or aggregation is driven through such processes of mixing two separate viscous materials, yielding a gel.

Carlfors et al. (1991) fabricated what they believed to be a thermodynamically stable oil-in-water as a topical delivery material. Their system has no thermally-gelling response. In fact, they report that their system becomes bicontinuous at elevated temperatures – we see no such transition in our system (see our added SAXS data in the SI, for example). There is no rheology reported in this article, most probably because there is nothing remarkable which occurs in the rheology. They assumed that the system was a microemulsion if the system was clear, homogenous and qualitatively low-viscosity (apparently by the eye). They report that some systems were checked to be in equilibrium by centrifugation, though it is not clear which and how many. In contrast, we show in our work that the order of mixing components (having the same final composition) gives rise to 2 different sets of metastable nanoemulsions with differing diameters (see new SI section “Optimizing the oil droplet size in the nanoemulsion”) - these are not equilibrium systems. Further discussion and data supporting that our system is a nanoemulsion is given in the new SI section “Verification of the formed nanoemulsion and its stability “.

> The reference to Carlfors et al. simply shows how far back in time the method of droplet production similar to the authors' extends and is labeled as 'microemulsion'; it was not intended as

an example of a rheologically interesting system. In other words, the way of creating a droplet system similar to that of the authors is decades old. Order of addition differences would be useful information for those interested in microemulsion formation, yet this does not make the system a nanoemulsion. Nanoemulsion formation by external flow-driving is different than microemulsion formation by what amounts to thermodynamic self-assembly.

Could a high molecular weight non-polar oil that is very strictly insoluble in water be used in the same process as the authors' to make nanoscale droplets? The authors do not demonstrate this. The molecular weight of the one type of emollient (C17 'oil') used by the authors for the droplet material is relatively low. Moreover, this emollient is more polar than a simple n-alkane, so this will further enhance its quantitative solubility in polar water. It is well known that hexadecane is an oil that is not very strictly insoluble in water near standard conditions, even though it is immiscible with water, and hexadecane's residual solubility in water can lead to many effects, including Ostwald ripening (see e.g. J. Weiss et al. 1999 Langmuir). Non-polar n-alkane oils having C-numbers around this range have been used in many kinds of microemulsion formulations for decades. So, the authors are controlling the rheology of a microemulsion system by using continuous-phase polymeric additives, just as Negi et al. enhanced their microemulsions with a continuous-phase hydrogel (that is known to have a temperature-dependent response) and obtained a soft-material gel that included microemulsion droplets.

We respectfully disagree with the reviewer's comment on controlling the rheology of our nanoemulsion by simply adding nanoemulsion to a continuous phase which would otherwise gel on its own (as was done by Negi et al.). Also, the rheological properties in the presented system can be controlled by multiple parameters, including oil fraction (Fig. S10), oil droplet size (Figure 4), Pluronic concentration (Fig. S11), Pluronic type (Fig. S7), and PEG concentration (Figure 5) – these are all supportive of the synergistic gelation mechanism we proposed.

> The reviewer previously understood about the 'synergistic' effect when writing the prior report. The prior comment was written to emphasize that a hydrogel containing nanoscale droplets formed as a microemulsion is known in the world of drug delivery, even if the polymer composition and mechanism of gelation could be different. This impacts the novelty of the authors' work. A comparative study would be needed by the authors to show that the authors' composition and structure would somehow perform better in drug delivery applications than the pre-existing composition and structure of e.g. Negi et al 2016 and others that have been published in the drug delivery literature since this publication, regardless of what may be synergistic effects in the mechanism of gelation in authors' system.

We have used different oils to prepare nanoemulsions, including oleic acid, squalene, ethyl laurate, and caprylic capric (composition is similar to our canonical formulation). However, in most cases the oil droplets size obtained via the PIC emulsification process resulted in a relatively large droplet diameter. For example, a nanoemulsion using squalene oil (chemical formula of squalene is C₃₀H₅₀, considered to be very hydrophobic oil) resulted in oil droplet size ~ 260 nm (PDI: 0.35) without a strong thermogelling behavior. However, using ethyl laurate as an oil phase resulted in a hydrodynamic oil droplet of ~ 60 nm, which forms a gel at an elevated temperature. As we also indicated in the manuscript (Figure 4), we speculate that the thermogelling behavior is highly correlated with the oil droplet size.

> Squalene can be nano-emulsified to sub-100 nm diameters readily using published nanoemulsification methods without requiring a great reduction in surface tension as is achieved via PIC. In fact, PIC places restrictions on the droplet compositions and sizes that could be better handled using other methods of nanoemulsification than a route that involves microemulsion formation via PIC.

While temperature-dependent gel-like materials can be made just of microemulsion phases themselves, the authors rely on adding pluronic block-copolymers to gel the system. The notion of gelling a system containing polymer components that associate differently at different temperatures has been established decades ago in the field of rheology, so the kind of temperature dependent behavior that the authors report is not surprising. The authors have not provided any direct imaging to substantiate their claim that a portion of the pluronic is actually in a

large fraction of droplets; their experimental evidence for this hypothesized mechanism is indirect and therefore not very convincing.

The reviewer does not fully appreciate a key point of the manuscript- gelation in our system does not occur solely due to the added Pluronics, nor with only the nanoemulsions – it is the synergistic effect of the two components which gives rise to the rich physics seen. This is new, not intuitive, and has not been established in the literature.

> The authors appear to be reading the above statement and inserting additional ideas that were not written in it. Nothing in the above statement implies that only the pluronic block-copolymers that the authors use would create a gel on their own without the droplets. While the exact system of the authors has not been established in the literature, at least the droplet portion largely has been regardless of nomenclature. Also, the rheology literature has other examples of colloid-polymer mixtures that gel when mixed yet neither system would be elastic on its own-- i.e. 'synergistic' effects (see e.g. very interesting flow-history dependent 'shake-gels' of laponite clay and PEO, just as a recent example). So, the notion of 'synergy' affecting the rheology of colloid-polymer mixtures and creating interesting rheological effects is not new. Combined together, these reduce the level of novelty.

Regarding direct imaging of the small oligomers onto the surface of the oil droplet – this is not feasible in our system. The scale of this problem, namely a ~50 nm liquid droplet interface combined with a small oligomer, makes it unfeasible to directly image the process. We do not have knowledge of a technique which would allow for imaging at this length scale and on liquid interfaces.

That said, we did provide quantitative and carefully constructed micro-DSC experiments to validate the adsorption of the copolymers onto the oil droplet interfaces. The micro-DSC captures the endo- or exo-thermic interactions between components in the solution as a function of temperature. This is a well-established technique used to study molecular-level adsorption and association processes. For example, other researchers investigated the interaction between Pluronic and nonionic surfactants using micro-DSC analysis (DOI: 10.1021/la0104267), or interaction between Pluronic and ionic surfactants (DOI: 10.1021/jp012729f), or effect of salt on aqueous solution properties of Pluronic (DOI: 10.1021/la9703712).

In addition to the micro-DSC data, our hypothesis is supported by the rheology experiments (a new discussion was added for the Fig. S3). The concentration of the copolymer used in this work is much lower than that to just simply gel the continuous phase. We demonstrate this in our control rheology experiments (Fig. S3).

The reviewer's comment did make us reflect on what direct imaging we could carry out. Confocal microscopy (which is the only practical imaging technique for our system) was utilized to image the microstructure of our nanoemulsion. The newly obtained results were added in the SI file (Fig. S4) and main text (highlighted text in page 6).

> These additional data and information do strengthen the authors' body of work on this system, even if they do not provide direct imaging evidence to compare with their hypothesis. Having direct imaging would significantly improve the novelty level of their work, if they can somehow find a way.

Whether or not the polymer used by the authors remains entirely in the continuous phase or partially enters the droplets would not really matter very much if the polymer material is mostly in the continuous phase and is imparting the dominant rheological characteristics of the soft system. This is especially true at a relatively low droplet volume fraction of 0.24. Unless the rheological properties of the system could be controlled over a dramatically larger range (e.g. several orders of magnitude higher in modulus) at a volume fraction of 0.24 than at a volume fraction of e.g. 0.02 for identical polymer concentration in the continuous phase, temperature, etc., then there isn't much in this study that would be interesting from a rheological point of view. The absence of results related to droplet volume fraction dependence makes one think that the polymer additive(s) are effectively controlling the rheology of the system, and if that is the case, then the

rheology of many pluronic solutions have already been well characterized as a function of concentration and temperature.

The reviewer is correct in noting that it is important to show that the oil phase plays a critical role in determining the rheological response in the system and that it is not just the polymer controlling the rheology. Indeed, we did show this in our control experiments in which we removed the oil and only looked at the rheology of the remaining polymers – this data is shown in the SI Fig. S3. Comparing this data to the main text Fig. 2, we note that indeed the elastic modulus changes by > 4 orders of magnitude by adding a small amount (volume fraction of 0.24) of oil droplets to the system. This data not only supports our mechanism, but clearly differentiates our work from the prior literature in which the gelling continuous phase was agnostic to the presence of the oil nanoemulsions.

We thank the reviewer for the suggestion of varying the oil fraction. We performed new experiments and added the results in Fig. S10 (different oil fraction). We also varied the gelator concentration and the new results are shown in Fig. S11. For example, G' is $\sim 4,000$ Pa at an oil fraction of 0.18, and increases to 10,000 Pa when oil fraction is ~ 0.35 .

> These are all good new results and do improve the quality of what is being presented.

Other groups have clearly made water-based gel-like soft materials containing nanoscale droplets of isopropyl myristate with polymer additives (e.g. hydrogel formers), so this material concept has already been devised and published, even if there is a difference in the authors' choice of gelling additive. Many hydrogels have strongly temperature dependent rheological behavior, so this temperature-dependent aspect could be anticipated based on the prior use of hydrogels in very similar formulations. These prior materials used by other groups also are biocompatible, suitable for drug delivery or other applications. Thus, the novelty level and wow-factor do not meet high standard of Nature Communications. The authors' work, after appropriate revision to address the existing literature that captures the central idea in their introduction and to put their results in context of this prior literature, would be most appropriate in a field-specific journal, related to a field such as soft matter or rheology.

We respectfully disagree with the main point above, namely we know of no other prior studies which show thermal gelation of nanoemulsions through the synergistic effects described in our article. The reviewer is correct that others have put nanoemulsions into continuous phases to form thermally gelling formulations, however in the prior work the continuous phase would have thermally gelled in of itself regardless of the nanoemulsion – our system will not. We have added one more reference in the article to clarify this point (ref. # 37). New physics were discovered in our work showing a subtle synergy of the two components (Pluronics and nanoemulsions) and a system developed which has potential applications due the choice of FDA approved components and a low energy formation method.

> Hydrogels containing nanoscale oil droplets, regardless of the method of formation, for the purpose of drug delivery using suitable components have already been introduced and published, so this very broad concept for drug delivery is not novel. The route of forming nanoscale droplets, regardless of nomenclature 'nanoemulsion' or 'microemulsion', used by the authors is effectively the same in published work on drug delivery application involving hydrogels. Both of these facts, regardless of 'synergistic effects' affect the novelty of the authors' work and were not apparent in their original submission. Their revised submission partially ameliorates these aspects, since at least this prior work is now cited, yet the revision retains most of its original wording rather than cleanly coming through with an initial paragraph or two that more appropriately sets the stage for their work using the most relevant references. Readers should be informed of the basic facts about the field and most relevant work before the authors then assert what is new, interesting, and different in their work. Readers should also know that synergistic gelation effects in the field of rheology of mixtures of small molecules and ions, colloids, and polymers in the same continuous phase are far from new, even if such synergistic effects have not been demonstrated previously for this particular mixture. The authors do not show comprehensive and comparative kinetic data that differentiate the delivery performance of their system over other existing systems that have compositions consisting of a combination of a hydrogel and nanoscale droplets, irrespective of the

gelation mechanism. So, the story from a drug delivery perspective is weakened by this. Perhaps the mechanism that leads to the gelation hypothesized by the authors actually negatively impacts the delivery aspects, compared to other systems in which the droplets are not incorporated as directly into networks, as is speculated by the authors for their system. The lack of direct visualization of the networks also means that additional real-space imaging information that could support their hypothesis of the gelation mechanism is not available.

So, while the addition of appropriate references and extra supporting evidence (e.g. through DSC, additional rheological measurements) represent favorable and significant improvements, the main issue with the revised manuscript still lies with its relatively low degree of novelty, at least in terms of criteria used by Nature Communications. Certainly, the exact system and results have not been published elsewhere. Yet, to those familiar with soft nanomaterials, rheology, and drug delivery, key structural data supporting the mechanism and a comparative assessment of the drug delivery potential, demonstrating superiority, are lacking. So, the novelty level still does not reach that required by the guidelines of Nature Communications.

Reviewer #3 (Remarks to the Author):

The authors have addressed my questions and comments by performing additional experiments (SAXS/CLSM), improving their control rheological experiments, and fully explaining the role of each ingredient to support the gelling mechanism. They have also addressed the other reviewers' comments based on a thorough understanding of nanoemulsion gelation mechanisms. Based on their response and the broad impact on academic and industrial investigations of emulsion systems, I am in favor of publication in NC.